# Gypsum heterogenous nucleation pathways regulated by surface functional groups and hydrophobicity

Yan-Fang Guan[1,6], Xiang-Yu Hong[2,6], Vasiliki Karanikola[3], Zhangxin Wang [4], Weiyi Pan [5], Heng-An Wu [2], Feng-Chao Wang [2] ✉, Han-Qing Yu [1] ✉ & Menachem Elimelech [5] ✉

Gypsum ($CaSO_4 \cdot 2H_2O$) plays a critical role in numerous natural and industrial processes. Nevertheless, the underlying mechanisms governing the formation of gypsum crystals on surfaces with diverse chemical properties remain poorly understood due to a lack of sufficient temporal-spatial resolution. Herein, we use in situ microscopy to investigate the real-time gypsum nucleation on self-assembled monolayers (SAMs) terminated with $-CH_3$, −hybrid (a combination of $NH_2$ and COOH), −COOH, $-SO_3$, $-NH_3$, and −OH functional groups. We report that the rate of gypsum formation is regulated by the surface functional groups and hydrophobicity, in the order of $-CH_3 > -$hybrid $> -$COOH $> -SO_3 \approx -NH_3 > -OH$. Results based on classical nucleation theory and molecular dynamics simulations reveal that nucleation pathways for hydrophilic surfaces involve surface-induced nucleation, with ion adsorption sites (i.e., functional groups) serving as anchors to facilitate the growth of vertically oriented clusters. Conversely, hydrophobic surfaces involve bulk nucleation with ions near the surface that coalesce into larger horizontal clusters. These findings provide new insights into the spatial and temporal characteristics of gypsum formation on various surfaces and highlight the significance of surface functional groups and hydrophobicity in governing gypsum formation mechanisms, while also acknowledging the possibility of alternative nucleation pathways due to the limitations of experimental techniques.

Gypsum ($CaSO_4 \cdot 2H_2O$) scale formation is important across diverse fields, impacting industrial processes and environmental systems[1–7]. One example is seawater desalination, where gypsum formation on reverse osmosis (RO) membrane surfaces leads to a decline in process performance and increased energy consumption[8–11]. In another example, deposition of gypsum on pore surfaces in underground oil reservoirs causes formation damage, which significantly hampers oil recovery efficiency[6]. A prevailing challenge in managing gypsum scaling is the lack of mechanistic understanding of heterogeneous scale formation on engineered surfaces. Therefore, unraveling the mechanisms of heterogeneous gypsum nucleation is crucial for informing the design of engineered surfaces and developing strategies to mitigate gypsum scaling.

[1]CAS Key Laboratory of Urban Pollutant Conversion, Department of Environmental Engineering, University of Science & Technology of China, Hefei, China. [2]CAS Key Laboratory of Mechanical Behavior and Design of Materials, Department of Modern Mechanics, University of Science and Technology of China, Hefei, China. [3]Department of Chemical and Environmental Engineering, University of Arizona, Tucson, AZ, USA. [4]Institute of Environmental and Ecological Engineering, Guangdong University of Technology, Guangzhou, Guangdong, China. [5]Department of Civil and Environmental Engineering, Rice University, Houston, TX, USA. [6]These authors contributed equally: Yan-Fang Guan, Xiang-Yu Hong. ✉e-mail: wangfc@ustc.edu.cn; hqyu@ustc.edu.cn; menachem.elimelech@rice.edu

Established over a century and a half ago, classical nucleation theory remains a foundational framework for characterizing nucleation tendencies from supersaturated solutions. This theory is broadly applicable to various substances, including calcite, gypsum, ice, proteins, and beyond[5,12–15]. According to classical nucleation theory, nucleation is a process that depends on the competition of nuclei between unfavorable surface energies and favorable bulk-free energies[4]. Several studies observed the effects of surface chemical properties on gypsum formation[5,16–18]. For instance, increasing surface hydrophilicity and creating superhydrophobic surfaces can potentially delay the formation of gypsum scale[16,17,19]. Other studies reported the importance of surface carboxylic groups, which could interact with $Ca^{2+}$ ions and promote the surface nucleation of gypsum[20–23]. In contrast, surface hydroxyl groups were found to have minimal or no effect on the formation of gypsum[20,22]. While these findings highlight the importance of surface chemical properties, the variability in results implies an intricate relationship between gypsum formation and surface properties. Hence, there is a crucial need to develop a comprehensive mechanistic understanding of gypsum nucleation and growth on surfaces with diverse functional groups, charge density, and levels of hydrophilicity.

Surface properties could also affect the initial stage of gypsum crystallization, which is beyond the description of classical nucleation theory. Recent studies with advanced characterization techniques, such as in situ X-ray small- and wide-angle scattering[15], high-resolution microscopy[24], and time-resolved cryogenic transmission electron microscopy[25], have observed stable precursor clusters or nanocrystalline at the early stage of nucleation, which would regulate the nucleation and growth behavior. Nonetheless, these stable pre-nucleation clusters were predominantly observed in the context of homogeneous nucleation processes, showing distinct kinetics compared to heterogeneous nucleation. Moreover, understanding of the underlying mechanism of this phenomenon is limited by the constraints of experimental observations, particularly the limitations of time and the quenching step.

Computer-based simulations—including density functional theory (DFT) calculations and molecular dynamics (MD) simulations—are powerful techniques for providing in-depth insights into the early stage of the nucleation process[3,7,26]. The mechanistic insights gained at the atomic and molecular levels could further be used to corroborate experimental results. For example, a combination of experiments and MD simulations revealed that classical nucleation theory can effectively elucidate the intricate mechanism governing calcite crystallization[26]. In addition to MD simulations, DFT calculations provide atomic-level details that enhance our understanding of the nucleation process[27–29]. However, only a few simulation studies have specifically addressed gypsum nucleation, particularly with respect to heterogeneous nucleation[30,31]. Consequently, a thorough investigation of the initial stages of gypsum nucleation through computational simulations is clearly warranted.

In this work, we investigated the role of surface properties in gypsum scale formation via systematic experimental observations and MD simulations. Self-assembled monolayers (SAMs) of alkyl thiols on gold surfaces were employed to create surfaces terminated with −NH₂, −OH, −COOH, −CH₃, −SO₃, and −hybrid (a combination of NH₂ and COOH) functional groups. An in situ imaging technique was employed to quantify the number of gypsum crystallites forming on these surfaces, enabling the calculation of the nucleation rate. Subsequently, we compared the observed gypsum nucleation rate with the number of cluster ions in the proximity of the surfaces as determined by MD simulations. Our results revealed that classical theories for crystal nucleation and growth combined with MD simulations can provide insights into gypsum nucleation mechanisms on surfaces with different chemical properties. Our findings also led us to propose two distinct gypsum growth mechanisms for hydrophilic and hydrophobic

surfaces. On hydrophilic surfaces, gypsum tends to grow vertically, influenced by the attractive induction interactions of adsorbed ions, whereas horizontal growth of gypsum prevails on hydrophobic surfaces.

## Results

### Fabrication and properties of different functionalized surfaces

Substrate surfaces with different functional groups were fabricated using SAM. Atomic force microscopy (AFM) showed that the surfaces were relatively smooth, with a mean arithmetic roughness below 1.3 nm (Supplementary Fig. 1). Surface hydrophilicity was assessed via water contact angle measurements with DI water as the probing liquid. As illustrated in Fig. 1, most substrate surfaces were hydrophilic, except for the methyl (−CH₃) functionalized surface which exhibits the highest static contact angle (98.10 ± 3.19°). The water contact angles for substrate surfaces terminated with a hybrid of NH₂ and COOH, −NH₂, −OH, −COOH, and −SO₃ groups were 81.83 ± 2.34°, 67.44 ± 6.54°, 60.78 ± 4.26°, 50.50 ± 8.03°, and 32.50 ± 1.78°, respectively.

X-ray photoelectron spectroscopy (XPS) was employed to analyze the elemental composition and oxidation states of functionalized substrate surfaces (Fig. 1), thereby verifying the successful modification achieved with a variety of function groups. For the −OH functionalized surface, the O1s peaks at 527.6, 529.7, and 531.1 eV were attributed to Au-O or Si-O, originating from the gold-coated silica wafer[32], while another peak at 533.3 eV was likely associated with C-OH in 11-mercapto-1-undecanol[33]. For the −COOH functionalized surface, three distinct components were revealed through deconvolution of the C 1s spectrum. The first component, detected at 284.8 eV, was attributed to C-C bonds within the aliphatic chain. The second component, observed at 286.6 eV, corresponded to the C-O/C=O bonds associated with the 11-mercaptoundecanoic acid. Lastly, the peak at 288.6 eV originated from the O-C=O bonds within the carboxylic group itself[34,35]. For the −NH₂ functionalized surface, two peaks at 400.0 and 402.2 eV were observed, corresponding to the neutral and protonated N-C species of amine tail groups, respectively[36]. The −CH₃ functionalized surface was primarily composed of C-H bonds (284.8 eV), attributed to the aliphatic chain (−CH₂ or −CH₃) of the 1-dodecanethiol molecule[35]. A negligible peak at 287.4 eV was observed, which may be attributed to slightly oxidized carbonaceous atmospheric contaminations, i.e., -C(O)-.[37]. The −SO₃ functionalized surface exhibited two peaks at 160.2 and 165.6 eV, arising from the gold substrate (Au-S) and sulfonic acid (-SO₃Na) groups, respectively[38]. The C 1 s signal was also examined on the hybrid surface and three peaks were observed. The peak centered at 284.8 eV was due to the C-C/C-H bonds of the molecule used. The second peak, shifted by 1.0 eV to higher energy (at 285.8 eV), was ascribed to C-N bond of the amide group. The third peak, located at 287.8 eV, represents either the C=O or C=O-N groups of the amide group[39]. The signal from N 1 s spectrum is at 400.0 eV (Supplementary Fig. 2), which is attributable to the neutral N-C species. Taken together, these results verify the successful binding of various groups onto the substrate surface.

### Nucleation kinetics of gypsum

The influence of different surface functional groups on the formation of crystallites on the substrate was examined through the application of an in situ imaging technique (Fig. 2A). To accelerate nucleation, solutions of CaCl₂ and Na₂SO₄ with a saturation index ($\sigma$) ranging from 0.97 to 1.63 were employed for gypsum formation. In all the nucleation experiments, a consistent solution flow rate of 30 mL h⁻¹ was used to the substrates functionalized with various functional groups. The nucleation and growth of gypsum crystallites (Supplementary Fig. 3) on the substrate surface were monitored over time via an optical microscope. The number density of gypsum crystallites was determined as a function of time for various $\sigma$ values, demonstrating a linear relationship where the number density of gypsum crystallites

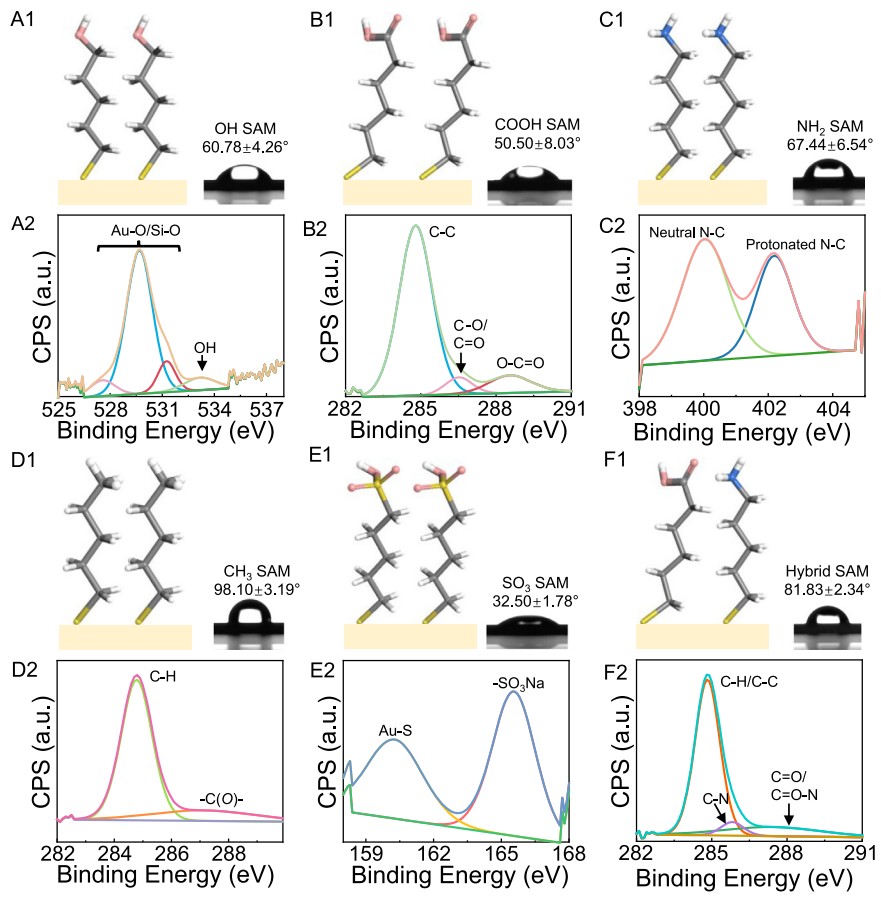

**Fig. 1 | Characteristics of SAMs.** The molecular structure and contact angles of (A1) −OH, (B1)−COOH, (C1)−NH₂, (D1)−CH₃, (E1)−SO₃, and (F1) hybrid of NH₂ and COOH groups terminated self-assembled monolayers. Pink, while, grey, yellow, and blue spheres represent oxygen, hydrogen, carbon, sulfur, nitrogen atoms, respectively. The XPS spectra of representative chemical bonds of (A2)−OH, (B2)−COOH, (C2) −NH₂, (D2)−CH₃, (E2)−SO₃, and (F2) hybrid of NH₂ and COOH groups terminated self-assembled monolayers. CPS and a.u. represent counts per second and arbitrary unit, respectively.

increased with time for all the collected data. Each type of substrate exhibited a distinct relationship (Fig. 2B and Supplementary Fig. 4). The steady-state rate of heterogeneous nucleation ($J_O$) was calculated from the slope of the number density vs. time. As expected, $J_O$ was proportional to the saturation index of the solution.

To quantify the thermodynamic barrier for heterogeneous nucleation of the different substrates, we plotted the relationship between $J_O$ and $\sigma$ (Fig. 2C). The slopes were calculated using Eq. 6 and were different for all substrates (Table 1). In particular, the −CH₃ functionalized substrate had the lowest $B$ value. Since $B$ is a representation of the thermodynamic barrier to nucleation, a lower $B$ value implies a more favorable nucleation. This finding indicates that substrates functionalized with −CH₃ groups (or hydrophobic surfaces) displayed a greater propensity for heterogenous nucleation of gypsum, in agreement with previous observations[1,2]. For the remaining functionalized substrates with more hydrophilic groups, the sequence of the heterogenous nucleation rates follows the order of hybrid > −COOH > −SO₃ ≈ −NH₃ > −OH surfaces. Interestingly, a higher nucleation rate was observed on the hybrid substrate terminated with −NH₃/−COOH than on the −NH₃ or −COOH terminated surfaces, suggesting the hybrid substrate was more amenable to gypsum nucleation than the purely aminated or carboxylated substrates. Prior investigations involving diatom biosilica and glass sponges have demonstrated that synergistic interactions between oppositely charged groups can effectively facilitate nucleation and mineralization processes[40,41]. In contrast to the −COOH terminated substrate surface, the −SO₃ and −NH₃ surfaces did not create favorable conditions to induce gypsum nucleation. The high nucleation rate on the substrate with −COOH

groups could be attributed to the strong complexation reaction between carboxylic groups and Ca²⁺, which promotes heterogeneous nucleation on the surface[20,22,23,42]. The substrate surface terminated with −OH groups exhibited the lowest gypsum crystallites count, indicating negligible interaction between hydroxyl groups and gypsum[20,22].

To elucidate the impact of distinct functional groups on nucleation rate from a molecular perspective, MD simulations were conducted to examine the incipient stages of gypsum nucleation and CaSO₄ growth on six different surfaces. At the beginning of simulations, calcium and sulfate ions are dispersed randomly in solution, predominantly existing as highly hydrated free ions. As the simulations progress, these free ions frequently collide with each other due to electrostatic attractive forces, resulting in the formation of precursor clusters. Monitoring the temporal evolution of free ions and precursor clusters enables us to categorize the nucleation process into two distinct phases. Taking the −CH₃ surface as an example, during the initial phase, the rapid decay of free ions coincides with the increase of precursor clusters at 5 ns (Fig. 3A), which manifest as chains, branches, and rings, consistent with prior studies on calcium sulfate nucleation[43,44]. These clusters subsequently evolve into larger aggregates over time (Supplementary Fig. 5), accompanied by a notable decrease in the number of free ions (Fig. 3A). Notably, the sizes of the three largest clusters undergo significant expansion in proximity to 5 ns (Supplementary Fig. 6). Therefore, the inflection point at about 5 ns could be indicative of a critical point, i.e., indicative of the critical size of nuclei. These findings are consistent with the classical nucleation theory, which suggests the existence of a critical size that

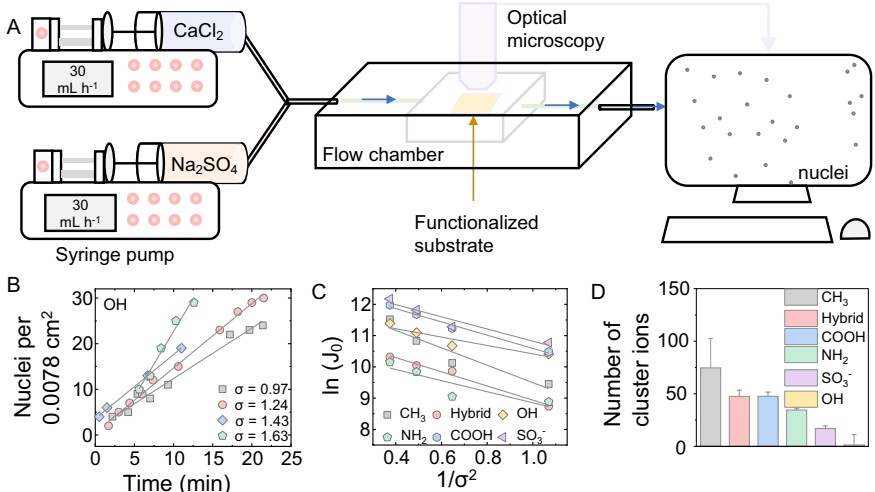

**Fig. 2 | Gypsum nucleation experiments. A** Schematic illustration of gypsum nucleation experiments. **B** The representative number density of gypsum crystallites versus time for various saturation index ($\sigma$) values on surfaces terminated with −OH. The slopes of lines for each supersaturation value represent the steady-state nucleation rate, $J_0$. Similar plots for surfaces terminated with −COOH, −NH₂, −CH₃, −SO₃, and hybrid of NH₂ and COOH groups are shown in Supplementary Fig. 4. **C** Gypsum nucleation rates obey classical nucleation theory; the linear relationship is predicted by Eq. 6 for all surfaces. The slope of the plots, $B$, represents the substrate-specific thermodynamic barrier to nucleation and is used to estimate gypsum–substrate interfacial free energy, $\gamma_{cs}$. **D** The total cluster ions near the surfaces (i.e., the distance between the center of mass (COM) of cluster minus the cluster radius and the surface groups is less than 15 Å) terminated with different functional groups as obtained from the MD simulations. Error bars are derived from the standard deviation of the average values calculated over the last 5 ns.

categorizes clusters as either subcritical or supercritical[45,46]. Simultaneously, the hydration degree of calcium ions diminishes and is replaced by coordination with sulfate ions (Fig. 3B). Hence, it can be inferred that a reduction in the number of free ions within the aqueous solution corresponds to a heightened degree of gypsum nucleation. In other words, a greater abundance of cluster ions signifies more pronounced nucleation. Moreover, the maximum cluster size serves as a suitable proxy for nucleation levels, as smaller precursor clusters tend to coalesce into larger ones.

While the nucleation stages of gypsum in solution remain remarkably similar (Fig. 3A and Supplementary Fig. 7), the nucleation degree varies across different surfaces (Supplementary Fig. 8). To assess the impact of surface groups on gypsum nucleation, only clusters near the surface (i.e., where the distance between the clusters center of mass (COM) minus its size and the surface groups is less than 15 Å) were calculated for comparison. The quantity of cluster ions and the maximum cluster size, which are crucial attributes governing the nucleation process and are challenging to ascertain experimentally[47], were tallied. Figure 2D reveals the highest concentration of cluster ions in the vicinity of −CH₃ surface, followed by the order of −hybrid > −COOH > −NH₂ > −SO₃ > −OH surfaces, in accordance with experimental findings. The observed trend in the maximal cluster size near the surface of various substrates (Supplementary Fig. 9) aligns consistently with this result. The findings from this study, along with recent investigations on calcite and calcium phosphate nucleation, offer compelling evidence supporting the effectiveness of classical nucleation theory in predicting the intricate nucleation rate on diverse surfaces with distinct chemical properties[26,48].

## Mechanisms of gypsum nucleation

Molecular dynamics simulations were employed to gain deeper insights into the heterogeneous nucleation of gypsum on diverse substrates. Based on classical nucleation theory, heterogeneous nucleation of a crystal requires nucleation of a solid−liquid phase boundary on a foreign surface[49–52]. For instance, calcium and sulfate ions can interact with the hydrophilic polar groups on the surface. However, further nucleation of calcium and sulfate ions to form CaSO₄ clusters is impeded on the hydrophilic surface due to the adhered water layers. On the contrary, hydrophobic surfaces could potentially facilitate the nucleation of calcium and sulfate ions to form CaSO₄ clusters[2]. Regardless of whether it occurs during the early pre-nucleation or later crystallization stages of CaSO₄, the process involves the displacement of the CaSO₄ to the hydration layer on the functionalized surfaces. As a result, the competitive interaction strength between water and CaSO₄ clusters with the surface plays a critical role in gypsum nucleation on the surface[16,18,53]. This competitive interaction strength can be quantitively described as the debonding work of CaSO₄, which describes the energy necessary to displace water by CaSO₄ at the substrate-water interface[54]:

$$W_{\text{debonding\_CaSO}_4} = (\Delta E_{\text{CaSO}_4/\text{surface}} + \Delta E_{\text{CaSO}_4/\text{water}} - \Delta E_{\text{water/surface}})/A \quad (1)$$

where $W_{\text{debonding\_CaSO4}}$, $\Delta E_{\text{CaSO4/surface}}$, $\Delta E_{\text{CaSO4/water}}$, $\Delta E_{\text{water/surface}}$, and $A$ represent the debonding work, the interaction energy between CaSO₄ and the surface, the interaction energy between CaSO₄ and water, the interaction energy between water and the surface, and the contact area of the interface, respectively. The molecular models employed for calculating interaction energies are illustrated in Supplementary Fig. 10. A negative $W_{\text{debonding\_CaSO4}}$ value suggests a spontaneous debonding process accompanied by energy release, while a positive value suggests the system requires external energy. Consequently, a more negative $W_{\text{debonding\_CaSO4}}$ value indicates that

## Table 1 | Average values of $B$ and ln($A$) determined from rates of gypsum nucleation from solution with varied levels of saturation

| Substrate | $B$ | ln($A$) |
|---|---|---|
| COOH | −2.12 | 12.71 |
| OH | −1.34 | 11.75 |
| NH₂ | −1.77 | 10.63 |
| CH₃ | −2.79 | 12.28 |
| SO₃ | −1.95 | 12.77 |
| COOH/NH₂ | −2.26 | 11.20 |

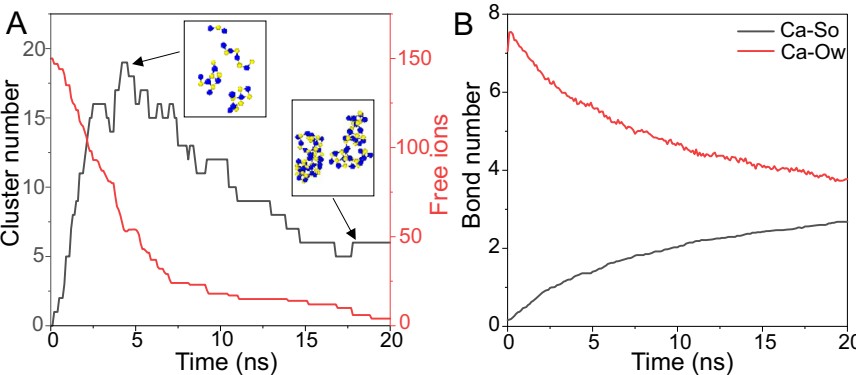

**Fig. 3 | The temporal evolution of free ions, clusters, and bond number. A** Time evolution of cluster number and free ions, taking -CH₃ surface as an example for analysis. The insets are the potential structures (chains, branches, and rings) of precursor clusters at the first stage, and the large clusters with an amorphous structure at the second stage of prenucleation process. **B** Time evolution of Ca-So and Ca-Ow bond numbers, which represent the coordination number of Ca²⁺ ions with SO₄²⁻ ions and water molecules, respectively.

$CaSO_4$ is more capable of displacing water at the substrate-water interface. The $W_{\text{debonding\_CaSO4}}$ values for the five substrate-water systems are all negative, which implies that each debonding process occurs spontaneously. The magnitude of debonding work follows a sequence of −hybrid > −COOH > −NH₂ > −OH > −CH₃ surfaces (Fig. 4A). The outcome of this debonding work on hydrophilic surfaces aligns with earlier cluster analyses and experimental findings. Notably, the debonding work magnitude on the hydrophobic −CH₃ surface is the smallest, implying the presence of an alternative nucleation mechanism.

In addition to debonding work, the energy ratios (ER) of CaSO₄-surface interfaces were also calculated to quantify the influence of water layer on CaSO₄-surface interfaces[54]:

$$ER = W_{\text{adhesion}} / W_{\text{debonding\_water}} \qquad (2)$$

where $W_{\text{adhesion}}$ is the adhesion work of the CaSO₄-surface interface (Supplementary Fig. 11A), and $W_{\text{debonding\_water}}$ is the debonding work of water (Supplementary Fig. 11B).

A higher ER value indicates that the CaSO₄-surface interface is less vulnerable to the water layer. As illustrated in Fig. 4B, the ER value of CaSO₄-surface follows a sequence of −hybrid > −COOH > −NH₂ ≈ −OH > −CH₃ surfaces. This finding indicates that the CaSO₄-hybrid surface interface is the least susceptible to the water layer, whereas the CaSO₄-CH₃ surface interface displays the highest susceptibility. Both the debonding work and ER values collectively indicate that the competitive interaction strength between ions and water with surface groups plays a crucial role in gypsum nucleation on hydrophilic surfaces. Conversely, gypsum nucleation on hydrophobic surfaces may involve an alternative mechanism.

Due to the weak interaction between ions and −CH₃ groups, it is presumed that ions in proximity to the surface can move more freely, thereby enhancing the probability of ion collisions that lead to the formation of larger clusters. Consequently, −CH₃ and −hybrid surfaces, representing hydrophobic and hydrophilic surface groups, respectively, were selected for comparing the standard deviation (SD) of potential energy at equivalent distances from the surface groups. In general, a greater disparity in potential energy corresponds to a heightened energy barrier that adsorbed ions must overcome when traversing the surface. As illustrated in Fig. 5A, the SD of potential energy for the −CH₃ surface indicates a substantially lower energy barrier for ion movement comparing with that of the −hybrid surface, as demonstrated in Supplementary Fig. 12. Moreover, the horizontal self-diffusion coefficient ($D_{xy}$) and mean-square displacement ($MSD_{xy}$) of CaSO₄ pairs on −CH₃ and −hybrid surfaces were calculated to quantitatively characterize the diffusion capability of surface ions. Specifically, two pairs of CaSO₄, initially positioned in close proximity to the surface and submerged in the aqueous solution with an approximate separation of 20 Å between them, were monitored for their diffusion behavior.

The horizontal self-diffusion coefficient of the hydrophobic −CH₃ surface is significantly greater than that of the hydrophilic hybrid surface, as illustrated in Fig. 5B and Supplementary Fig. 13. This difference indicates a superior ion diffusion capacity on the −CH₃ surface and a heightened likelihood for small CaSO₄ clusters to expand into larger clusters in a horizontal direction. This observation is further supported by the trajectory lines of two CaSO₄ pairs above −CH₃ and −hybrid surfaces (the inset figures (i) and (ii) of Fig. 5B). When considering both scenarios, we observe two distinct nucleation pathways for CaSO₄ clusters on hydrophobic and hydrophilic surfaces. On hydrophobic surfaces, CaSO₄ clusters exhibit a weaker affinity to the surface and tend to aggregate into larger, planar clusters within the bulk solution. Conversely, CaSO₄ clusters exhibit a stronger adsorption capacity on hydrophilic surfaces, leading to multisite nucleation. This multisite nucleation encourages ions to form upright clusters oriented vertically to the surface, as illustrated in Fig. 5C. It is worth noting that unstable upright clusters, situated at a single adsorption site, are more susceptible to displacement by dynamic water flow compared to the flat clusters with a larger surface contact area.

In contrast, prior research has demonstrated that ions exhibit a higher affinity for adsorption on hydrophilic surfaces as opposed to hydrophobic ones, potentially resulting in ion enrichment at the interface between the substrate and water[52,55,56]. Through careful examination of ion distribution along the direction perpendicular to the substrates, we observed a distinctive pattern: on hydrophilic surfaces, the ion count decreases as distance increases, while on the hydrophobic −CH₃ surface, the opposite trend emerges (Fig. 5D). This finding suggests that more ions are directly adsorbed on hydrophilic surfaces compared to the hydrophobic −CH₃ surface. Nevertheless, when summing the ion count within a distance of less than 25 Å, the cumulative ion number for the hydrophobic −CH₃ surface is the largest (Supplementary Fig. 14). This observation provides additional support for the notion that gypsum nucleation on the hydrophobic −CH₃ surface primarily takes place through bulk nucleation rather than being induced by ion adsorption.

For hydrophilic and charged surfaces, the adsorption of more counter ions on the surface can promote nucleation[1,16,53]. However, it has been documented that an excessive net charge can trigger electrostatic repulsion when the interfacial ion concentration reaches extremely high levels[57,58]. Consequently, an alternative pre-adsorbed

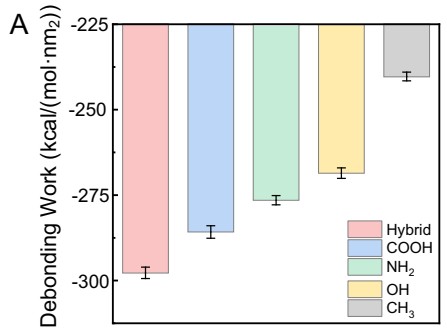
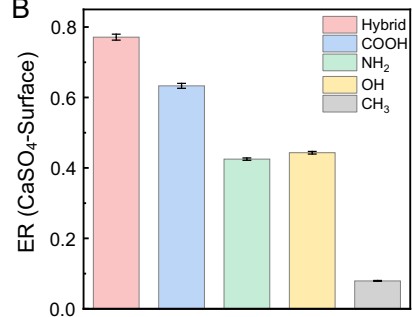

**Fig. 4 | Effect of adhered water layers on CaSO₄ nucleation. A** The debonding work of CaSO₄ displacing water from surfaces terminated with different functional groups. **B** The energy ratios (ER) of between CaSO₄ and surfaces terminated with different functional groups. ER is obtained from the adhesion work of the CaSO₄-surface interface divided by the debonding work of water. Error bars are derived from the standard deviation of the average values calculated over the last 5 ns.

ion layer model was constructed. As a representation of hydrophilic surfaces, the −hybrid surface was selected, and an ion layer with varying quantities was deliberately positioned upon it. Subsequently, a small CaSO₄ probe containing several calcium and sulfate ions approached the surface from a considerable distance, enabling the computation of the total force exerted by the CaSO₄ probe, as illustrated in Fig. 5E. The pre-adsorbed ion layer model, devoid of water molecules, was employed to assess the impact of ion adsorption on the surface deposition dynamics of CaSO₄. Notably, the maximum adhesion force of the CaSO₄ probe shows a non-monotonic trend as ion concentration increases, first rising and then declining. Considering the attributes of the Lennard-Jones (LJ) potential, it becomes evident that as atoms come into close proximity, the prevailing interaction transitions from attraction (over long distances) to repulsion (at shorter distances). Thus, a repulsion distance was defined to describe the range of the repulsive force. This distance was calculated by subtracting the zero-crossing point of the surface groups' density profile from the force zero-crossing point of the CaSO₄ probe. The repulsion distance expands as the number of ions increases (Fig. 5F). This result suggests that an excess of ions adsorbed on the surface hinders further ion adsorption, thereby shedding light on why gypsum nucleation on hydrophobic surfaces is, to some extent, more pronounced than on hydrophilic surfaces.

The above results identify two nucleation pathways for hydrophilic and hydrophobic surfaces: surface-induced nucleation with vertical growth and bulk nucleation with horizontal growth, as illustrated in Fig. 6. The first pathway is initiated by the adsorption of ions onto surfaces and typically manifests as multisite nucleation. In this process, ion adsorption sites serve as pivotal anchors, facilitating the formation of clusters that grow vertically, and perpendicular to the surface. In contrast, the second pathway unfolds when the interaction between ions and the surface is relatively weak. This permits bulk ions near the surface to collide and fuse into larger clusters. On hydrophobic surfaces, where robust adsorption sites are lacking, clusters tend to expand horizontally, adopting a flat configuration that can subsequently accumulate on the surface.

## Discussion

The intricate interplay between gypsum and a foreign surface distinguishes this process from the non-classical nucleation model, which is widely considered the most prevalent mechanism for gypsum nucleation in homogeneous supersaturated solutions[15,24,43]. For example, a previous study using in situ small-angle X-ray scattering revealed that gypsum nucleation initiates from the aggregation of sub-3 nm primary species or CaSO₄ precursor[15]. Other in situ techniques, including high-resolution transmission electron microscopy (TEM), cryo-TEM, and scanning electron microscopy (SEM), have provided valuable insights into homogeneous nucleation

processes[15,24,43,59]. However, these techniques may face challenges in fully capturing in situ heterogeneous nucleation due to the complex interplay between nuclei and foreign surfaces under hydrated conditions.

Compared to gypsum nucleation in bulk solutions, the interactions between gypsum and surfaces in heterogeneous nucleation render the process more complex and less understood, particularly regarding the relationship between the kinetics and structures of critical gypsum nuclei. Here, we employed in situ microscopic observations to directly investigate gypsum formation under fully hydrated conditions on these surfaces. This approach allows us to observe the well-ordered water molecules around functionalized surfaces, which play a crucial role in influencing the thermodynamics of nucleation and growth, as supported by previous studies[60,61]. Such a focus on heterogeneous nucleation provides a different perspective from the conditions studied in the aforementioned literature. Our experimental and simulation results corroborate that gypsum nucleation aligns with classical nucleation theory. On hydrophilic surfaces, gypsum nucleation is primarily induced by specific ion adsorption sites (i.e., functional groups). This induction leads to the growth of gypsum crystals oriented vertically. Moreover, variations in the rates of gypsum nucleation on distinct hydrophilic surfaces can be explained through their respective debonding energies. In contrast, hydrophobic surfaces exhibit weak interactions between surface functional groups and gypsum, resulting in bulk nucleation, low potential energy, and rapid self-diffusion in the horizontal direction.

Given that nucleation data can be fitted using CNT even when the pathway is non-classical[26], we recognize that fitting our data to CNT does not definitively imply a classical nucleation mechanism. To explore the possibility of an intermediate phase that may arise in non-classical pathways, we conducted surface plasmon resonance microscopy experiments, which allow for the nanoscale spatiotemporal identification of nuclei by accurately measuring the refractive index of individual nuclei without interference from background signals[62,63]. Since the refractive index of crystalline materials is often higher than that of amorphous or liquid materials[64,65], the phase transformation of CaSO₄ in the early nucleation process could be identified by this method, which would show a sudden signal change. By tracking the trajectories of CaSO₄ formed on the surface, a time-resolved plasmonic image sequence was captured (Supplementary Fig. 15a). Upon nucleation, a typical scattering pattern was clearly observed. As time progressed, the plasmonic intensity increased (Supplementary Fig. 15b), indicating CaSO₄ nuclei growth. Notably, the monotonically increasing plasmonic intensity suggests that an abrupt phase transition in CaSO₄ is unlikely. Additionally, we calculated the great changes in the RDF of sulfur atom pairs of sulfate with time. To identify the specific structure of the clusters, we compared the RDF diagrams of amorphous phases and the perfect crystal of CaSO₄, using ion

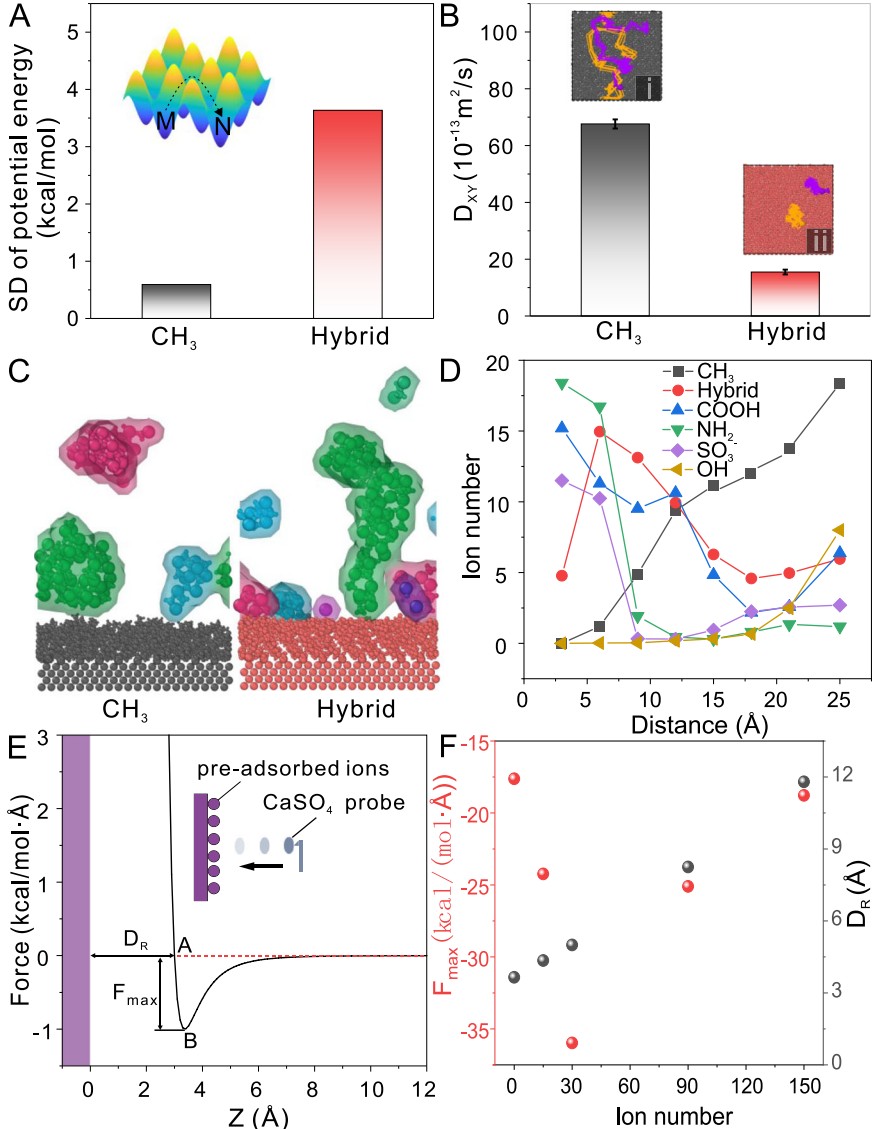

**Fig. 5 | Gypsum nucleation mechanism. A** Standard deviation (SD) of potential energy for surfaces terminated with $CH_3$ and hybrid functional groups. The inset is a schematic diagram depicting the potential energy surface. **B** Horizontal self-diffusion coefficient ($D_{xy}$) of $CaSO_4$ pairs on surfaces terminated with $CH_3$ and hybrid functional groups. The inset i and ii are the trajectory lines of two $CaSO_4$ pairs above the surfaces terminated with $CH_3$ and hybrid functional groups, respectively. Each bar in Fig. 5b is the mean ± s.d. ($n = 3$). **C** Representative simulation snapshots of $CaSO_4$ clusters on $-CH_3$ and $-$hybrid surfaces. The ions are colored to distinguish different $CaSO_4$ clusters and the water molecules were removed for clarification. **D** Ion distribution with the distance along the direction perpzndicular to the surfaces with different functional groups. **E** Schematic diagram on the calculation of the maximum adhesion force ($F_{max}$) and repulsion distance ($D_R$) of $CaSO_4$ probe. **F** The maximum attraction force ($F_{max}$) and repulsion distance ($D_R$) of $CaSO_4$ probe approaching to the surface with pre-adsorbed ions.

concentrations within the typical supersaturation range for homogeneous nucleation. As illustrated in Supplementary Fig. 16, the RDF peaks of amorphous calcium sulfate appeared broad and flattened, while the RDF peaks of crystalline calcium sulfate were sharp and well-defined, indicating a degree of long-range order. Simulations performed for the clusters over the period from 0 to 50 ns suggest that the behavior of the clusters was similar to that of a crystal phase. Therefore, no direct evidence of an intermediate phase was observed.

Previous studies observed that the fundamental concepts of classical nucleation theory remain applicable for calcite, iron (hydr) oxide, and ice nucleation on foreign surface[13,66–69]. A non-classical nucleation process has also been found in surface-induced nucleation[1,2]. It is important to note that the preference for nucleation on either hydrophilic or hydrophobic surfaces can vary significantly based on the particular mineral system and experimental conditions in play. Factors such as supersaturation, temperature, the presence of impurities, and the surface functional groups play crucial roles in the nucleation process. Additionally, several simulations have suggested that pathway selection could be modulated by adjusting the interaction between nuclei and the foreign surface[13,70]. In this work, steady-state nucleation experiments with supersaturated solutions were conducted on different foreign surfaces to ensure the successful application of classical concepts.

Contrary to previously proposed scenarios[1,18], we conclude that surface free energy is insufficient for predicting gypsum nucleation on different surfaces. When plotting the surface energy of each substrate against their respective $B$ values (i.e., thermodynamic barriers to nucleation), no robust correlation is observed (Supplementary Fig. 17). Furthermore, surface free energy measurements of hydrophilic surfaces revealed only marginal differences, which contradicted earlier assumptions that surfaces with higher negative charges would result in accelerated nucleation rates[10,17,20,22,23].

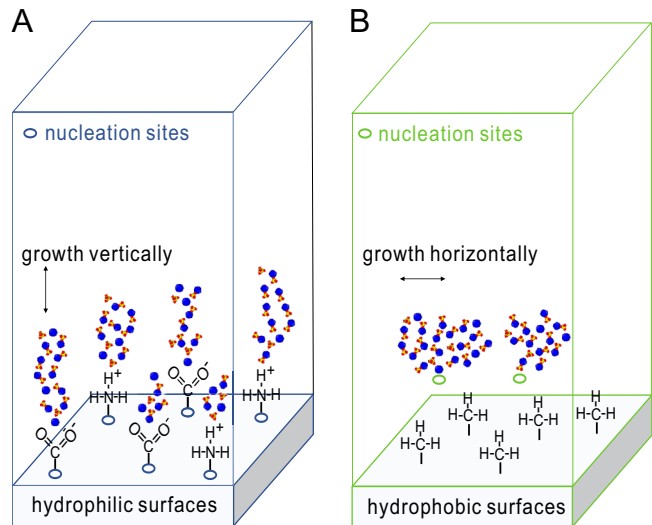

**Fig. 6 | Nucleation pathways.** Schematic diagram illustrating gypsum nucleation on hydrophilic and hydrophobic surfaces.

Revisiting the concept of interfacial free energy ($\gamma$) within classical nucleation theory, it signifies the collective energy of the crystal-liquid-substrate system and is presented in Eq. S4 in the Supplementary Note. 2. A large $\gamma_{SC}$ (crystal-substrate) coupled with a small $\gamma_{SL}$ (substrate-liquid) contributes to a high interfacial free energy. Consequently, nucleation rates hinge upon the competition between the energetics involved in gypsum nuclei formation on the substrate and the establishment of a new substrate-nuclei interface, as opposed to the energetics of water displacement near the substrate, where water is attracted. However, it is important to note that the measured surface free energy, as determined by DLVO theory, pertains to the energy associated with the substrate-air interface. In this study, we calculate the debonding work, which quantitatively describes the energy needed to displace water by the $CaSO_4$ cluster on the substrate-water interface. This debonding work aligns well with the observed nucleation rate on hydrophilic surfaces.

This study takes a significant stride in understanding the gypsum nucleation mechanism through a combination of experimental observations and MD simulations. Our findings align with CNT, providing a foundation for a more comprehensive and precise grasp of heterogeneous gypsum nucleation on surfaces, a critical aspect in the effective management of mineral scaling in various industrial processes (e.g., membrane desalination). Nonetheless, it is important to recognize the limitations of our experimental techniques—specifically, the inability to directly observe critical cluster sizes, dissolved clusters, and crystalline structures that remain internally amorphous. It is possible that certain non-classical processes may have evaded our detection. Consequently, deeper exploration into the early stages of gypsum's complex growth patterns on diverse surfaces, using advanced characterization techniques operating at the sub-nano scale or even smaller, will be more convincible to comprehensively capture these potential alternative pathways.

## Methods

### Preparation and characterization of functionalized surfaces
Substrates for gypsum nucleation were prepared by first coating silicon wafers with a 5-nm thick titanium (99.995%, Kurt J. Lesker Co., USA) layer followed by coating with an 80-nm gold (99.999%, Kurt J. Lesker Co., USA) layer via a thin film deposition system (EJ1800, Kurt J. Lesker Co., USA). Prior to the functionalization of the substrates, the surfaces were cleaned and activated using the approach developed by Biolin Scientific[71]. The cleaning process involved the removal of

potential organic contaminates on the substrate surfaces by exposing them to UV-ozone for 10 min, and then submerging the surfaces in a 5:1:1 mixture of ultrapure DI water, 25% v/v ammonium hydroxide, and 30% v/v hydrogen peroxide for 15 min at a temperature of 70–75 °C. After cleaning, the substrates were rinsed with ultrapure DI water and dried with nitrogen gas. The cleaned substrates were then immersed in ethanol-based, 1.5 mM solutions of six different alkanethiols from Sigma-Aldrich: 11-mercaptoundecanoic acid, 1-dodecanethiol, sodium 2-mercaptoethanesulfonate, 11-amino-1-undecanethiol, hydrochloride, 11-mercapto-1-undecanol, and a mixture of 11-mercaptoundecanoic acid and 11-amino-1-undecanethiol, hydrochloride. After 24 h, the functionalized substrates were rinsed with ethanol and dried with nitrogen gas.

The affinity of water to the substrate surface was determined by measuring the water contact angle utilizing the sessile-drop method with a goniometer (Attension-Biolin Scientific Inc., USA)[9]. For each substrate surface, water contact angles were measured at eight different locations on each substrate surface, and the obtained data was analyzed using post processing software (VCA Optima XE, AST Products Inc., USA). Substrate surface chemical composition and valence states were analyzed by X-ray photoelectron spectroscopy (XPS, *VersaProbe* II, PHI Inc., USA) using monochromatic Al Kα radiation and with a 0.47 eV system resolution. The binding energy was calibrated with C 1 s peaks at 284.8 eV. Peak fitting and data analysis were conducted using Casa-XPS software. Substrate surface roughness was evaluated using atomic force microscopy (Dimension Icon, Bruker Co., USA). Surface scans (5 μm × 5 μm) were probed at 5 random locations on each sample, and the arithmetic mean roughness (Ra) was calculated. Surface plasmon resonance microscopy experiments were conducted on sensing chips modified with a self-assembled monolayer using a commercial inverted microscope (Ti microscope, Nikon, Japan), equipped with a 100× oil-immersion objective lens (NA = 1.49). This setup provides exceptional sensitivity for detecting variations in interface refractive indices, allowing for real-time, label-free detection of dynamic processes in solution[63]. Illumination was achieved by a 660 nm superluminescent diode to excite surface plasmon resonance[62]. Plasmonic images were captured by a CCD camera (Pike-032B, Allied Vision Technologies) through a 0.46 × zoom-out lens (Nikon, Japan)[62].

### Nucleation rate measurements
To isolate the surface-induced heterogeneous nucleation process, experiments were performed under conditions where no homogeneous nucleation in the bulk solution takes place. In these experiments, calcium chloride and sodium sulfate solutions were prepared and transferred into two separate polypropylene 60-mL syringes. The two solutions were mixed at a 1:1 volume ratio and then introduced into a custom-built acrylic glass flow chamber. Inside the flow chamber, the substrate was placed in a space with a diameter of 25 mm and a depth of 1.5 mm. The flow rate was maintained at 30 mL h$^{-1}$. The experiments were conducted at room temperature (25 °C) at various solution saturation indices, ranging from 0.97 to 1.63 (Supplementary Table 1). The saturation index ($\sigma$) is defined as Eq. 3:

$$\sigma = \ln\left(\frac{\alpha_{Ca^{2+}} \times \alpha_{SO_4^{2-}}}{k_{sp}}\right) \tag{3}$$

where $\alpha$ is the ion activity, as calculated by PHREEQC software (version 3.0)[72], and $K_{sp}$ is solubility product for $Ca^{2+}$ and $SO_4^{2-}$ ($K_{sp} = 10^{-4.58}$)[73].

Gypsum nucleation on the functionalized substrate surface was monitored by optical microscopy (10× objective, Zeiss). We note that the resolution of the optical microscope was insufficient to directly observe the formation of gypsum crystallites on the substrate surface[5,18]. As a result, the nucleation rates we reported in this study were based on the assumption that each crystal observed from the

 

optical microscope was initiated from a single crystallite and all crystals were formed heterogeneously on the substrate surface. We calculated the nucleation rates based on the following key assumptions:

Each crystal observed optically was originated from a single nucleus. We assume that each nucleus overcame the energy barrier necessary to reach a critical size and subsequently grew into an optically observable crystallite. This assumption is substantiated by the data collected during the intervals when the spacing between crystallites was significantly large (Supplementary Fig. 3 A). During these intervals of steady nucleation, each crystal manifested as a distinct, well-faceted rhombohedrum. Moreover, the linear relationship of crystal appearance over time, as illustrated in Fig. 2 and Supplementary Fig. 4, supports the independence of nucleation events. The absence of such linearity would suggest interdependencies among nucleation events, which was not observed.

### Classical nucleation theory

Based on classical nucleation theory, the rate of crystallization on a substrate surface can be described by

$$J_0 = A \exp\left(\frac{-\Delta G}{k_B T}\right) \tag{4}$$

where $J_O$ is the steady-state rate of heterogeneous nucleation, $A$ is a kinetic pre-factor, which includes rates of ion attachment, diffusion, and desolvation, $k_B$ is the Boltzmann constant, $T$ is the absolute temperature, and $\Delta G$ is the free energy barrier to form a critically sized crystal. The latter is defined as

$$\Delta G = \frac{F\omega^2\gamma^3}{\sigma^2 k_B^2 T^2} \tag{5}$$

where $F$ is a constant that depends on the crystal shape factor, $\omega$ is the molecular volume of gypsum, $\gamma$ is interfacial energy of the crystal-substrate-liquid system, and $\sigma$ is the saturation index, defined earlier (Eq. 3).

Substituting Eq. 5 into 4 and rewriting to a linear form yields

$$\ln(J_0) = \ln(A) - B\left(\frac{1}{\sigma^2}\right) \tag{6}$$

where

$$B = \frac{F\omega^2\gamma^3}{k_B^3 T^3} \tag{7}$$

By determining the slope of $\ln(J_O)$ vs. $1/\sigma^2$, we can obtain $B$, which is proportional to the substrate-specific thermodynamic barriers to nucleation and is used to estimate gypsum–substrate interfacial free energies, $\gamma$.

### Molecular dynamics simulations

All molecular simulations were performed using the large-scale atomic/molecular massively parallel simulator (LAMMPS) package[74]. To accelerate nucleation events within a feasible simulation timescale and minimize computational costs, the simulation was performed at elevated supersaturation levels. The simulation models were constructed by placing a supersaturated calcium sulfate ($CaSO_4$) solution consisting of 9000 water molecules, 75 calcium ions, and 75 sulfate ions above the substrates modified with different functional groups. The size of the simulation box is $5.77$ nm $\times$ $5.77$ nm $\times$ $14.56$ nm, and the volume of the aqueous solution is about $2.7 \times 10^5$ nm³. There is a vacuum slab around 3 nm along the perpendicular direction to the substrate. Once the initial models were constructed, an energy

minimization was first performed to minimize the system's potential energy via the steepest descent algorithm. The simulation systems were then relaxed to simulate the nucleation process of gypsum in the NVT (isothermal-isovolume) ensemble at the temperature of 300 K for 20 ns. The last 5 ns are adopted for the data analysis. The time step for the Velocity-Verlet integration scheme is 1 fs. CHARMM force field[75] was used for the substrate, and single point charge/extended (SPC/E) model[76] was adopted for the water molecules. In addition, the interactions of calcium, sulfate, sodium, and chloride ions were described by widely used ion force field parameters[77–79]. The interactions among different types of atoms were computed by the Lorentz–Berthelot (LB) mixing rules. The inner and outer cutoff distances were set as 0.8 nm and 1.2 nm. The long-range Coulomb electrostatic interactions were treated by particle-particle-particle mesh (PPPM) algorithm. The OVITO software[80] was used for visualization.

### Cluster analysis of simulation data

Cluster analysis of the simulation data was based on the simulation snapshots taken at 0.1 ns intervals. We first defined the $CaSO_4$ cluster, which was an aggregation of $(Ca)_n(SO_4)_m$ ions where $(m + n)$ is larger than 2, by the first valley value of the radial distribution function (RDF) profile of Ca-S pairs. As shown in Supplementary Fig. 18, the $SO_4^{2-}$ and $Ca^{2+}$ are considered as a $CaSO_4$ cluster when the distance between Ca and S atoms is lower than 4.25 Å. Therefore, the calcium and sulfate ions in the simulation systems can be categorized as cluster ions and free ions. Although there are water molecules bound to ions, only the solute ion-pairs were considered in the cluster analysis. The cluster size is calculated as the total number $(n + m)$ of calcium and sulfate ions in the clusters. At each frame, the cluster number, cluster size, cluster ion number, and free ion number were recorded. The clusters were classified as the clusters near the surface and the clusters in the bulk solution according to their gyration radius and the relative position with surfaces along the Z direction. When the distance between the center of mass (COM) of clusters minus the cluster gyration radius and the surface is less than 15 Å, the clusters are thought to be the clusters near the surface; otherwise, the clusters are in the bulk solution. The position of the boundary line of the surface is determined by identifying the location at which the density of the terminal functional groups reaches half of its peak density.

## Data availability

The data supporting the findings of this work are available within the paper and its Supplementary Information files. All other relevant source data are available from the corresponding authors on request. Source data are provided with this paper.

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

## Acknowledgements

This work was supported by the National Natural Science Foundation of China under Award Nos. 52200025 (Y.F.G.), 51821006 (H.Q.Y.), 52192684 (H.Q.Y.), and 12241203 (F.C.W.), and also acknowledged in part by the National Science Foundation through the Engineering Research Center for Nanotechnology-Enabled Water Treatment under Award No. EEC-1449500 (M.E.). F.C.W. acknowledges the Youth Innovation Promotion Association CAS (2020449). This publication was developed under a graduate fellowship awarded to Y.F.G. by the China Scholarship Council (CSC). The numerical calculations in this work were conducted in the Supercomputing Center of University of Science and Technology of China.

## Author contributions

Y.F.G., H.Q.Y., and M.E. designed the research; Y.F.G. performed the research; X.Y.H., F.C.W., and H.A.W. conducted the molecular simulations; Y.F.G., X.Y.H., F.C.W., H.Q.Y., V.K., Z.X.W., and M.E. analyzed the results; and Y.F.G., X.Y.H., W.Y.P., H.Q.Y., and M.E. wrote the paper. All authors contributed to discussion of the results and the manuscript.

## Competing interests

The authors declare no competing interests.
