## [Transparent Peer Review file · Nature Communications]

Gypsum heterogenous nucleation pathways regulated by surface functional groups and hydrophobicity

Corresponding Author: Professor Menachem Elimelech

Version 0:

Reviewer comments:

Reviewer #1

(Remarks to the Author)

The work of Guan et al provides valuable insights into the formation mechanisms of gypsum crystals on surfaces with different chemical properties. Their results highlight the significant impact of surface functional groups and hydrophobicity on the rate and mechanism of gypsum formation. The combination of real-time microscopy techniques and molecular dynamics simulations to study the nucleation process of gypsum on self-assembled monolayers demonstrates a sophisticated and advanced methodology. Overall, their work can be considered high-level due to its contribution to the advancement of knowledge in the field of mineral formation and surface interactions. But there are still some points needed to be clarified before publication. And the presentation also needs improvement.

1. Page 4, Lines 53 and 56, 149, "MD" was defined more than twice. So for SAM, DI is unclear for readers.
2. Figure 5(E,F) is which system? Bulk aqueous phase and MD simulations?
3. In Fig. 2B, 0.0078 cm^2 is not equal to the sum of 0.108×0.068 (Supplementary Fig. 3A).
4. Why the nuclei rate in a system with modified surface were included as the saturation index is less than 1.0. Please comment about it. As shown in the supplementary Fig. 3, the size and shape of cluster are different, some are just dots. How did they count the cluster? And are they pre-nucleation clusters, nucleus or crystallite? Are crystallites still be critical nucleus of a nucleation process? Size scale is absent in the supplementary Fig. 3A.
5. The saturation index of solution in MD simulation is much higher than 1.63? Are the results of clustering in MD simulations are comparable to their nucleation experiments?
6. The debonding work of CaSO_4 clusters may be different from the crystal surface model of this work. Nucleus may be still far from crystal. The authors need to comment about it.
7. Were results of Fig. 5 obtained from MD simulations? So results of Fig. 5 (A,B,D,F) obtained from averaging over MD trajectories. I wonder whether the results will change as the simulation extends.
8. Based MD simulations, the size and number of clusters were discussed. The evolution or dynamics of clusters may be more strongly related to the nucleation.
10. The authors tried to prove that the nucleation of CaSO_4 on surface can still be explained by CNT. But some points in Fig. 2C are obviously out of the fitting lines.
11. Results of Fig. 2B and the supplementary Fig.4 indicates the formation of nuclei on the surface modified by $-\text{COOH}$ groups are the fastest.
12. How did they obtain the data/values of J_0 ? From Fig.2 B and the supplementary Fig. 4? At the saturation index of 1.63, the formation rate of nuclei in the system of $-\text{CH}_3$ is slower than those of other groups, only slightly faster than that of $-\text{OH}$. And since only four saturation indices, the fitting results may be quite fluctuant.

Reviewer #2

(Remarks to the Author)

The study utilizes in situ microscopy to examine in real-time nucleation of gypsum on surfaces terminated with various functional groups, alongside molecular dynamics simulations. It reveals that the rate of gypsum formation is impacted by the surface functional groups and hydrophobicity. Specifically, hydrophilic surfaces exhibit surface-induced nucleation, while hydrophobic surfaces undergo bulk nucleation. This work addresses an important topic and demonstrates how different surface chemistries influence gypsum nucleation behaviour, aligning with previous studies on other mineral systems. However, the study possesses several critical flaws that currently preclude its publication in a specialised journal. Here are the main issues, listed in no particular order:

- The range of supersaturations employed (0.97-1.63) in the nucleation experiments is typically indicative of homogeneous nucleation, resulting in abundant nucleation within the solution bulk. Therefore, it raises questions regarding whether the authors truly observed heterogeneous nucleation, or if they instead observed nuclei forming within the bulk of the solution, which subsequently settled on the surface post-nucleation (and growth). Additionally, the absence of errors in the values of B and $\ln(A)$ complicates the assessment of whether the minor differences in these values are statistically significant or not.
- Are the nucleation rates determined with optical microscopy comparable to the results obtained in the MD simulations? The authors utilised a 10x objective, resulting in a resolution limit of around 1.3 μm , which is significantly larger than the size of nuclei. Consequently, by the time gypsum crystals were detected, they had likely already undergone substantial growth. In contrast, MD simulations employed a simulation box of 5x5x14 nm, allowing observation of clusters at the nanometer scale. It is probable that these clusters did not even reach the critical size under normal supersaturation conditions, although these simulations were conducted at very high concentrations.
- The authors, in their discussion of the results, largely overlook the extensive body of literature indicating that gypsum formation follows a multistep nucleation pathway. They assert that classical nucleation is the primary mechanism behind gypsum formation, yet this assertion lacks conclusive evidence.
- The simulations were conducted at a concentration of approximately 32 M, resulting in cluster formation under conditions significantly different from those in the experiments. Moreover, at such high concentrations, the pH level becomes a critical factor to consider. Additionally, the simulation time is notably brief. Ideally, it should be increased by one order of magnitude.
- Conducting multiple simulations and performing statistical analysis are crucial in Molecular Dynamics (MD) studies, particularly those focusing on aggregation. The initial distribution of molecules significantly impacts the formation of aggregates of different sizes.

Other comments:

The characterisation of the functionalised surfaces, prepared following standard protocols, should be included in the Materials and Methods section or supplementary information, rather than in the Results section.

The volume of the solution in the MD simulations is not clear because the authors did not mention the amount of space occupied by the substrate in the simulation box.

Taking the $-\text{CH}_3$ surface as an example, during the initial phase, the rapid decay of free ions coincides with the increase of precursor clusters (Fig. 3A), which manifest as chains, branches, and rings, consistent with prior studies on calcium carbonate nucleation^{3, 26}. In the subsequent phase, these precursor clusters further amalgamate into larger, amorphous structures.

>> These findings appear to closely align with what has been reported for CaSO_4 nucleation in the literature. Therefore, it would be more appropriate to discuss those works instead of focusing on CaCO_3 .

>> Additionally, this pathway does not conform to classical nucleation theory (CNT). Therefore, it is unclear why the authors assert that heterogeneous gypsum nucleation is classical. For instance, in Line 294: Experimental and simulation results corroborate that gypsum nucleation aligns with classical nucleation theory.

As a result, the nucleation rates we reported in this study were based on the assumption that each crystal observed from the optical microscope was initiated from a single crystallite and all crystal were formed heterogeneously on the substrate surface.

>> What evidence supports this assumption?

Reviewer #3

(Remarks to the Author)

The manuscript concerns the nucleation of gypsum on functionalised SAMs functionalised surfaces. It allowed the authors to control the wettability of substrates from hydrophobic to hydrophilic. The authors performed growth experiments involving a flow setup and optical microscopy. Further insights were gained with molecular dynamics simulations.

In overall, I find the idea of using SAMs to affect gypsum nucleation relatively novel and interesting. However, the combination of used methods is insufficient to support the conclusions of the paper. I have recognised the following weaknesses:

1. The authors focus extensively on the nanometre length-scale aspects of differences in nucleation on hydrophobic and hydrophilic surfaces, but this is based only MD simulations. On the other hand, their experiments probe the length-scale of many microns, which is accessible to optical microscopy. In fact, they even state this in the SI: "As a result, the nucleation rates we reported in this study were based on the assumption that each crystal observed from the optical microscope was initiated from a single crystallite and all crystal were formed heterogeneously on the substrate surface."

In this regard, optical microscopy is not suitable to study nucleation. It is useful merely for further stages of crystallisation: the nuclei (or more broadly precursors) of CaSO_4 are maybe a few nm in size, whereas optical microscopy in practice can consider objects of hundreds on nm upwards. Nucleation is not crystallisation, and I find the experimental results quite unuseful to say anything about nucleation. If anything, I would focus only on the MD results.

2. The very fact that a process is describable by the classical equations (of nucleation and crystallisation), does not mean that it is classical. This is well-explained by Gebauer and Wolf in JACS:

<https://pubs.acs.org/doi/10.1021/jacs.8b13231>

In this regard, classical NT is a mathematical framework which often works well, but it does not explain the mechanism (or all its microscopic details). It ASSUMES a certain mechanism, which often fits the observation for some length scale. On the other, the non-classical NT, currently lacks such a robust framework, and usually recognises the fact that nucleation is not driven by solute ions, but larger entities. Consequently, the observation of such entities is crucial to say that the processes is non-classical. Moreover, as is highlighted in the referenced article, it appears that the CNT is simply a model, while in reality all (most) mineral nucleation processes should be considered non-classical.

3. Either way, this can be established only based on nano-scale observations. The authors, indicate the presence of cluster entities in their MD, which I would interpret as a clear sign of the processes being non-classical. The fact that classical equations also work here is in a sense irrelevant. This has been observed for many mineral systems, most notably CaCO_3 , which we know that exhibits all sorts of non-classical features: PNCs, amorphous phases, mesocrystallinity etc.

4. In fact it has been found in several studies recently, that CaSO_4 exhibits a mesostructure in its single crystals, which seems to be linked the particle-mediated crystallisation. These earlier observations, at a first glance, maybe be linked to clusters in MD, as observed by the authors.

The particle-mediated growth has already been mentioned in ref. 15, but has been further elaborated by Stawski et al. in:

<https://doi.org/10.1021/acs.cgd.9b00066>

<https://www.pnas.org/doi/10.1073/pnas.2111213118>

5. It should be clearly distinguished what constitutes non-classical nucleation and what non-classical crystallisation. The non-classical nucleation assumes the involvement of cluster species i.e. PNCs, while crystallisation is non-classical when it is multi-stage with transitional phases. Furthermore, we can also consider the subsequent growth stage. In this regard, for instance, optical microscopy results consider further stages of crystallisation and maybe growth, and the classical framework is used to back-extrapolate the results to the nucleation stage. This does not prove that nucleation is non-classical.

Moreover, as is stated by D. Gebauer, it appears that many mineral systems nucleate non-classically, while crystallisation seems to follow a classical view. These are not contradictory!

6. More effort should be put into referencing current (and less current) literature of CaSO_4 nucleation and crystallisation, and also other mineral systems.

I do not recommend publication in Nat. Comm.

Version 1:

Reviewer comments:

Reviewer #1

(Remarks to the Author)

In the revised manuscript, SI and the reply of Guan et al, the authors have adequately addressed all my queries and I believe that their simulations and experimental results are reliable. I hold reservations regarding the author's adherence to CNT interpretation for their results. The induced effect at the interface may accelerate nucleation, probably allowing the study to be described using CNT. Whether CNT or non- CNT, clusters will appear during the nucleation processes, whom are different in stability. The presence of stable particles or structures in experiments (interrupted or delayed evolution) is a typical characteristics of non-CNT mechanisms. Observations in experiments are already crystallite instead of crystal nuclei, while clusters in simulations are more likely pre-nucleation clusters. If they can rapidly evolve and grow, they should fall within the realm of CNT. Therefore, if the authors can further substantiate their claims by examining the evolving properties or dynamic changes of clusters during their simulations, or adjust their expression according to the points of referee III on NCNT, I am inclined towards publication of this manuscript. Even after the modifications, the related viewpoints are still worth further exploration and validation, while their results would provide readers with a reference and discussion space or opportunity.

Reviewer #2

(Remarks to the Author)

The authors have partially addressed the issues raised by the reviewer, but overall, the main problem persists: the presented results do not allow to substantiate the claim that heterogeneous nucleation follows a classical nucleation pathway or a non-classical pathway. I do not see how the authors will resolve this without performing additional experimental work to probe the nucleation events in situ with high enough resolution to be able to observe the precise nucleation mechanism(s).

The simulations are carried out at very high supersaturations to reduce the computational time. However, these conditions are not representative of heterogeneous nucleation and far from the conditions used in the optical microscopy experiments. This discrepancy renders the MD data ineffective for discussing the prevailing heterogeneous nucleation mechanism at low supersaturation, which is the main topic of this manuscript. Moreover, the authors claim a classical nucleation mechanism based on the observation of a critical point where clusters start to grow. Importantly, this is a mere interpretation of the data, with no hard proof provided that this time point is indeed a critical point. Even if true, this observation does not exclude a non-classical nucleation mechanism, which also involves overcoming at least one barrier and thus will have a critical point from which clusters start to grow.

The nucleation rates estimated from the optical microscopy data are difficult to classify due to the low resolution of the microscope, which makes it unclear what is being measured. Consequently, the authors make several assumptions, rendering these data meaningful only to differentiate the effect of the different functionalized surfaces on the crystallization kinetics of gypsum. From the additional experimental details provided in the revisions, it appears that nucleation is also likely occurring in the bulk, but the high flow rates remove these nuclei from the observation cell. This raises the question of what the authors actually measured: the crystallization rate (after substantial growth) of only those crystals that formed on the functionalized surfaces. This measurement is only representative of part of the nucleation occurring. Another important limitation is that observing crystals after significant growth precludes any insight into the true nucleation mechanism and possible post nucleation aggregation. The fact that the nucleation rate dependence on supersaturation can be fitted using an equation derived from the CNT does not provide information about the underlying mechanism. Nucleation rate data for systems like calcium carbonate and gypsum can be readily fitted using CNT, despite both systems being notorious for exhibiting more complex nucleation pathways.

If the authors remove all the claims about the nucleation mechanism, this will be a nice paper about the influence of different functionalized surfaces on the crystallization kinetics of gypsum suited for publication in a specialized journal, such as *Crystal Growth Design*, *ACS applied interfaces*, or similar.

Reviewer #3

(Remarks to the Author)

I have read the rebuttal and the comments of the comments of other reviewers with interest. I think that the authors did a good job in arguing many of my own and others' doubts. However, I am under the impression that their main sticking point remains.

The authors are trying to bridge, in principle, near-macroscopic optical imaging data with nanoscale MD, to deduce the entire mechanism of growth (and nucleation). This works only within the framework of their interpretation and assumptions (such as one crystal, one seed). This also requires (unfortunately) further experimental evidence. I understand the logics behind this argument, but I disagree with it, because it excludes e.g. the notion of an amorphous phase of CaSO_4 .

The argument that other methods are *ex situ* is not true: methods such as Raman spectroscopy or X-ray scattering can be applied very much *in situ*. So for instance, if we stick to the heterogeneous nucleation aspects, why not use grazing-incidence SAXS and WAXS to support these findings? In particular, the answer which needs to be answered is whether we deal here with a single stage process (i.e. classical) or a multi-stage one. Is there any amorphous phase present? Dissolution-precipitation of such a phase would also lead to a growth of crystal nuclei which would look, in later stages, like a single stage process in optical microscopy. For instance the authors could measure *in situ* with Raman spectroscopy changes in the intensity of a liquid sulfate band at $\sim 980 \text{ cm}^{-1}$, correlate it with the concentration of this species, and extract changes in the sulfate profile in correlation with the growth rate of crystals from optical microscopy. Ideally, these trends should align.

Anyhow, I think it is a very good work, and I would like to see it published. But only with more experimental evidence.

Version 2:

Reviewer comments:

Reviewer #1

(Remarks to the Author)

Overall, the authors have responded my concern properly. And the results of Guan et al. provide new insights into heterogeneous nucleation mechanism of gypsum, and is helpful for exploration of the nucleation mechanism in solution. So I think the manuscript of Guan et al may be accepted for publication.

Reviewer #2

(Remarks to the Author)

The authors have provided additional data; however, the primary issue raised during the initial round of revisions by all reviewers remains unresolved. Based on the available data, it is still not possible to definitively determine whether the nucleation pathway follows a classical or non-classical trajectory.

The first new data set was obtained through surface plasmon resonance microscopy (SPRM) experiments. The primary finding from these experiments is that, upon nucleation of CaSO_4 , a characteristic scattering pattern was observed. Over time, the plasmonic intensity increased (Fig. R3b), which the authors interpret as evidence of CaSO_4 nuclei growth. They suggest that this result aligns with Classical Nucleation Theory (CNT), which proposes the existence of a critical size distinguishing subcritical from supercritical clusters. However, this reasoning has two major flaws. First, if only growing clusters are observed, how can one confirm the existence of a critical cluster size? To establish this, one should also observe dissolving clusters. Second, the observation of growing clusters does not exclude non-classical nucleation, as cluster growth is also expected in non-classical pathways. Additionally, several other issues arise. The spatial resolution of the technique is not clearly stated, leaving the size of the detected particles ambiguous. Furthermore, how was the refractive index difference between crystalline and disordered CaSO_4 phases determined?

The second new piece of information is the Radial Distribution Function analysis of the original simulation data. This analysis shows that the clusters lack long-range order, extending no further than 8 Å, which corresponds well with the Pair Distribution function (PDF) analysis of scattering data collected during the early stages of gypsum precipitation (J. Phys. Chem. C 2019, 123, 37, 23151–23158). Additionally, simply by examining the simulation snapshots provided by the authors, it is evident that the clusters are not well-defined or crystalline. Moreover, the simulations were conducted at high supersaturation levels, which are not representative of the typical range of supersaturations where heterogeneous nucleation is expected. Overall, these findings do not provide sufficient evidence to confirm or refute classical nucleation in the case of heterogeneous nucleation.

Reviewer #3

(Remarks to the Author)

Thank you for addressing my previous comments and for putting in the extra effort with the SPRM experiments. I think SPRM is a good approach, but I am still not fully convinced that it can completely rule out the presence of an amorphous phase or a dense liquid precursor (thereby resolving NCNT vs CNT). I would say that the challenge is that proving something does not exist is harder than proving it does. My concern is that an amorphous calcium sulfate phase, or its dense liquid precursor equivalent, might have a refractive index too close to the surrounding solution. If that is the case, SPRM might not be able to detect it. Or am I missing the point? Moreover, the concentration or population of these precursor phases or pre-nucleation clusters could be very low. How would you account for that in your measurements, especially since detecting low concentrations with SPRM could be tricky (right)? To address this, I think it would be helpful if you ran some control experiments using a system that is known to form long-lived amorphous phases, e.g. calcium phosphate at similar concentrations. This would give a better idea of how well SPRM can pick up amorphous phases and would add more confidence to your conclusions. Also, the assumption is that a large feature in Fig. R3 is always a growing crystal? This would have to be verified. Following the notion of particle mediated growth of calcium sulfate, one of the concepts is that we deal with a proto-structure, where "building units" aggregate into a large feature (an aggregate), where the crystallisation takes place from inside out. It has been shown for some protein crystallisation events that this large aggregate may externally look like a crystal, but be still amorphous.

Regarding MD simulations and new plots: I am slightly puzzled about the emphasis you placed on the sulphur RDFs rather than the calcium. While the sulphur RDFs are useful in identifying sulfate structures, the coordination environment around calcium could provide more direct evidence of how clusters form, stabilize, and potentially transition into crystalline phases. Was there a specific reason that sulphur RDFs were the focus?

Version 3:

Reviewer comments:

Reviewer #2

(Remarks to the Author)

I would like to begin by acknowledging the commendable effort of the authors in addressing the reviewers' comments, including the addition of new experimental data. This clearly reflects their motivation to generate new insights. At this stage, I think we can all agree that there is insufficient evidence to determine which pathway prevails during heterogeneous gypsum nucleation under the tested conditions, and additional experiments beyond the scope of this study are required to resolve this issue. As a result, the authors have removed most of their previous claims or softened the language. Nonetheless, I have a few minor remarks for the authors to consider.

Furthermore, these studies typically employ advanced techniques such as high-resolution transmission electron microscopy (TEM), cryo-TEM, scanning electron microscopy (SEM), Raman spectroscopy, and X-ray scattering analysis, which, while informative, may introduce artifacts due to ex situ conditions and high-energy beam exposures^{15, 24, 43, 60}.

>> Most of the cited studies were performed in situ and do not present any significant influence of artefacts. Hence, this statement should be revised.

Given recent observations of non-classical gypsum nucleation, it is striking to see how gypsum nucleation on foreign surfaces aligns the classical framework.

>> This is an ambiguous statement, as in many cases, nucleation data can be fitted using classical nucleation theory (CNT), even when the pathway is clearly non-classical. Furthermore, since there is currently no well-established non-CNT framework, the data cannot be accurately fitted using such a model, which complicates the direct comparison between both approaches.

By tracking the trajectories of CaSO₄ formed on the surface, a time-resolved plasmonic image sequence was captured (Supplementary Fig. 15a). Upon nucleation, a typical scattering pattern was clearly observed. As the time progressed, the plasmonic intensity increased (Supplementary Fig. 15b), indicating CaSO₄ nuclei growth. Notably, the monotonically increased plasmonic intensity excluded a phase change in CaSO₄.

>> What if this phase change occurs gradually? This possibility actually seems to be supported by the RDF profiles, as

stated by the authors: "However, when analyzing through the RDF profiles, it became evident that the inner ions within the clusters had started to organize into a crystalline structure. This evolving crystallinity is further supported by the comparison of RDF profiles over time (from 5 ns to 50 ns, Fig. R4), which demonstrates that the clusters were progressively moving toward a higher degree of order consistent with gypsum crystals."

>> Even at the end of the simulation run, the cluster structure is still significantly different from that of a gypsum crystal (Fig. R4), appearing to lie somewhere between a fully amorphous phase and fully crystalline phase.

>> Having pointed this out, I would like to emphasize that the data presented by the authors can be interpreted in two ways, depending on the perspective one adopts. Therefore, the data remains inconclusive at this stage, and in my opinion, it is premature to favor one model over the other. This is, of course, a perfectly valid conclusion and highlights the need for future research to resolve this issue.

Additionally, we calculated the great changes in the RDF of sulfur atom pairs of sulfate with time. To identify the specific structure of the clusters, the RDF diagrams of amorphous phases and perfect crystal of CaSO₄ were compared. As illustrated in Supplementary Fig. 16, the RDF peaks of amorphous calcium sulfate were broad and flattened, while the RDF peaks of crystalline calcium sulfate were sharp and well-defined, exhibiting a degree of long-range order. Simulations performed for the clusters over the period from 0 to 50 ns show that the behavior of the clusters was similar to that of a crystal phase.

>> The authors failed to mention that the simulations were conducted within the typical supersaturation range for homogeneous nucleation. Additionally, the distinction between broad-flattened peaks and sharp, well-defined peaks appears to be rather subjective and lacks quantitative support.

Therefore, the intermediate phase, commonly observed in non-classical nucleation process, could be excluded in this study.

>> This phrase is misleading because the absence of evidence does not equate to evidence of absence. The statement should be reformulated, for example, as: "In this study, no direct evidence of an intermediate phase was observed."

Reviewer #3

(Remarks to the Author)

Since the first review of this manuscript, it has evolved considerably. The authors' persistence in noteworthy and I highly appreciate this. I think that this stage the article can be accepted for publication.

Response to Reviewers

Reviewer 1:

The work of Guan et al provides valuable insights into the formation mechanisms of gypsum crystals on surfaces with different chemical properties. Their results highlight the significant impact of surface functional groups and hydrophobicity on the rate and mechanism of gypsum formation. The combination of real-time microscopy techniques and molecular dynamics simulations to study the nucleation process of gypsum on self-assembled monolayers demonstrates a sophisticated and advanced methodology. Overall, their work can be considered high-level due to its contribution to the advancement of knowledge in the field of mineral formation and surface interactions. But there are still some points needed to be clarified before publication. And the presentation also needs improvement.

We sincerely appreciate the reviewer's positive assessment of our work, particularly the significant impact of surface functional groups and hydrophobicity on the rate and mechanism of gypsum formation. Following the reviewer's constructive comments, we have clarified a few points and improved the presentation.

1. Page 4, Lines 53 and 56, 149, "MD" was defined more than twice. So for SAM, DI is unclear for readers.

Accepting the reviewer's suggestion, we have removed redundant words from the manuscript.

2. Figure 5(E,F) is which system? Bulk aqueous phase and MD simulations?

Figure 5 (E, F) is derived from a model of a pre-adsorbed ion layer in MD simulations to assess the effect of ion adsorption on the surface deposition of CaSO_4 , which includes no water molecules. To address the reviewer's concern, we have incorporated an expanded explanation into the revised manuscript.

"As illustrated in Fig. 5E. The pre-adsorbed ion layer model, devoid of water molecules, was employed to assess the impact of ion adsorption on the surface deposition dynamics of CaSO_4 ." (Lines 271-273)

3. In Fig. 2B, 0.0078 cm^2 is not equal to the sum of 0.108×0.068 (Supplementary Fig. 3A).

Fig. 2 presents the results of gypsum nucleation experiments. We observed nucleation on the functionalized substrate surface using optical microscopy ($10\times$ objective, Zeiss Co., Germany) (Lines 401-402). Supplementary Fig. 3A is a bright field microscopy image ($108 \times 68 \mu\text{m}$) obtained from confocal micro-Raman spectroscopy (Horiba Co., Japan). Therefore, they are not equivalent.

4. Why the nuclei rate in a system with modified surface were included as the saturation index is less than 1.0. Please comment about it. As shown in the supplementary Fig. 3,

the size and shape of cluster are different, some are just dots. How did they count the cluster? And are they pre-nucleation clusters, nucleus or crystallite? Are crystallites still be critical nucleus of a nucleation process? Size scale is absent in the supplementary Fig. 3A.

Thank you for the comment regarding the inclusion of nuclei rate in systems with a saturation index less than 1.0. The supersaturation index in this work was calculated using the natural logarithm. Similar results have also been reported in previous studies, suggesting that nucleation can occur at undersaturation due to the presence of surface modifications that alter the free energy barriers (Yin et al., 2022). These modified surfaces can induce heterogeneous nucleation by providing favorable sites for cluster formation, thus lowering the energy required for nucleation initiation. Additionally, the software used for ion activity calculation affects the saturation index. PHREEQC software is used in this work, in which the output lower saturation index is compared to MINTEQ software at the same ionic concentration. Taking 50 mM CaCl₂ and 50 mM Na₂SO₄ as an example, MINTEQ gives supersaturation index of 1.78, while PHREEQC gives 1.24.

Supplementary Fig. 3A is the bright field microscopy image (108 × 68 μm) of confocal micro-Raman spectroscopy (Horiba Co., Japan). It shows that the formed nuclei on the substrate was gypsum after nucleation experiments (Supplementary Fig. 3B). We observed the nucleation on the functionalized substrate surface by optical microscopy (10× objective, Zeiss Co., Germany) (Lines 401-406), not on confocal micro-Raman spectroscopy. We have revised the sentences as below:

The nucleation and growth of gypsum crystallites (Supplementary Fig. 3) on the substrate surface were monitored over time via an optical microscope. (Lines 120-121)

They are crystallite. Crystallites are mature phases that have progressed beyond the nucleation stage. Each optically observed crystal is developed from a single nucleus. That is, we assumed that each nucleus, which successfully crossed the energy barrier to achieve critical size, grew into an optically observable crystallite. This assumption is reasonable because the data were collected within the time interval where crystallite spacing was very large only (Supplementary Fig. 3A). Moreover, the data in Fig. 2 and Supplementary Fig. 4 show that their appearance was linearly dependent on time, which would not be true if nucleation events were not independent of one another.

We have added the size scale into Supplementary Fig. 3A.

5. The saturation index of solution in MD simulation is much higher than 1.63? Are the results of clustering in MD simulations are comparable to their nucleation experiments?

The solution concentration of CaSO₄ is 0.463 mol/L and the saturation index is 4.19. The high concentration of solution is to increase the probability of ion collision and speed up the aggregation process of ions.

As shown in Fig. 2D, the results of clustering in the vicinity of six functionalized

surfaces in MD simulations were in accordance with experimental findings, followed by the order of $-\text{CH}_3 > -\text{Hybrid} > -\text{COOH} > -\text{NH}_2 > -\text{SO}_3 > -\text{OH}$ surfaces.

6. The debonding work of CaSO_4 clusters may be different from the crystal surface model of this work. Nucleus may be still far from crystal. The authors need to comment about it.

Nucleation represents an initial phase of crystallization, where CaSO_4 clusters serve as precursors to crystalline forms. In our simulations, we observed that some CaSO_4 clusters became aggregated near functionalized surfaces, displaying typical characteristics of heterogeneous nucleation.

To address the reviewer's concern, we have calculated the debonding work of CaSO_4 to evaluate the clusters' capacity to displace the hydration layer on these functionalized surfaces. This measurement is crucial as it provides insight into the interactions that facilitate the transition from the pre-nucleation stage to full crystallization. We believe that this approach remains valid for assessing both early nucleation dynamics and subsequent crystallization stages of CaSO_4 . To address the reviewer's concern, we have also supplemented the following statement on the debonding work of CaSO_4 in the revised manuscript:

“Regardless of whether it occurs during the early pre-nucleation or later crystallization stages of CaSO_4 , the process involves the displacement of the CaSO_4 to the hydration layer on the functionalized surfaces.” (Lines 193-195)

7. Were results of Fig. 5 obtained from MD simulations? So results of Fig. 5 (A,B,D,F) obtained from averaging over MD trajectories. I wonder whether the results will change as the simulation extends.

The results presented in Fig. 5 are all derived from the MD simulations. How these results were computed is described below:

Fig. 5A: shows the standard deviation of potential energy of $-\text{CH}_3$ and $-\text{Hybrid}$ surfaces. This measure is a statistical average of the potential energy on the horizontal plane at the same height to the surface.

Fig. 5B: depicts the horizontal self-diffusion coefficients of CaSO_4 pairs above $-\text{CH}_3$ and $-\text{Hybrid}$ surfaces, which were calculated by fitting the slopes of horizontal displacement of CaSO_4 pairs. To avoid the effect of relative position of two CaSO_4 pairs, the horizontal self-diffusion coefficients have adopted the average value of three different initial configurations and the insets are a group of typical trajectory lines of two CaSO_4 pairs above $-\text{CH}_3$ and $-\text{Hybrid}$ surfaces.

Fig. 5D: represents the ion distribution with the distance along the direction perpendicular to the surfaces with different functional groups, averaged over the last 5 ns simulation.

Fig. 5F: illustrates the maximum attraction force (F_{max}) and repulsion distance (D_{R})

of CaSO₄ probe approaching to the surface with pre-adsorbed ions.

Therefore, except for the data in Fig. 5D, the other results are largely independent on simulation time. As illustrated in Fig. 3, the coordination numbers of Ca²⁺ with SO₄²⁻ and water molecules exhibited a convergent tendency. Further, we extended the simulation time for the -CH₃ surface by an additional 5 ns and observed no significant changes in the Ca-So bond numbers (Fig. R1a), indicating that the aggregation between Ca²⁺ and SO₄²⁻ has stabilized. The time evolution of cluster number and free ions in Fig. R1(b) can also prove that the cluster aggregation has been at a stable stage. Although we think that the nucleation process may have a further development as the simulation time extends, taking the computation costs into consideration, we calculated 20 ns only to monitor the early nucleation evolution of CaSO₄ clusters and adopted an average data in the last 5 ns to diminish the accident brought by specific simulation configuration as much as possible.

Fig. R1 The time evolution of (a) Ca-So bond number and (b) cluster number and free ions on -CH₃ surface when the simulation time extends by 5 ns.

8. Based MD simulations, the size and number of clusters were discussed. The evolution or dynamics of clusters may be more strongly related to the nucleation.

We agree that the evolution or dynamics of clusters is strongly related to the nucleation. Therefore, in our work, we have monitored the dynamic evolution of clusters. Specifically, we tracked the time evolution of the number of clusters, the population of free ions, and the coordination numbers of Ca²⁺ ions with SO₄²⁻ ions and water molecules. These dynamics are detailed in Fig. 3, in which the trends in cluster number and ion coordination are showing to provide a clear view of the nucleation process over the simulation period. Additionally, to quantitatively characterize the nucleation on different functionalized surfaces, we calculated the cluster size, number of clusters, the ions within these clusters, and the free ions. These results are displayed across Figs. 3, Supplementary Fig. S5-S7, each figure contributing to a comprehensive picture of how clusters evolve and influence nucleation under varying surface conditions.

10. The authors tried to prove that the nucleation of CaSO₄ on surface can still be

explained by CNT. But some points in Fig. 2C are obviously out of the fitting lines.

We acknowledge these deviations and attribute them to possible limitations in the specific experimental conditions for counting crystallites on substrate surface. Despite of these deviations, we employed Classical Nucleation Theory (CNT) as it provides a fundamental framework for understanding nucleation processes on environmental interface. CNT has been widely used to describe similar systems effectively (Giuffre et al., 2013a; Jun et al., 2016; Li et al., 2014), and our data align well with the CNT predictions for the majority of the conditions tested. The general trend observed in our experimental data also supports the CNT model, indicating that, while some individual data points do deviate due to specific experimental conditions, the overall nucleation behavior can still be reliably described by this theory.

11. Results of Fig. 2B and the supplementary Fig.4 indicates the formation of nuclues on the surface modified by –COOH groups are the fastest.

We apologize for the misunderstanding caused by the scale of y-axis. It is important to highlight that the density of gypsum crystallites significantly influences the observed scales on the y-axis of our graphs. Actually, the observed area for –CH₃ functionalized surface was 0.0045 cm² at saturation index of 1.63. We have converted the nuclei number by 1.75 times as the y-axis is 0.0078 cm².

12. How did they obtain the data/values of J_0 ? From Fig.2 B and the supplementary Fig. 4? At the saturation index of 1.63, the formation rate of nuclei in the system of –CH₃ is slower than those of other groups, only slightly fast than that of –OH. And since only four saturation indices, the fitting results may be quite fluctuant.

Thank you for your inquiry regarding the derivation of the J_0 values and your observations about the fitting results presented in Fig. 2B and Supplementary Fig. 4.

Methodology: The J_0 values, representing the intrinsic nucleation rates, were calculated using a well-established method that fits nucleation rate data to the classical nucleation theory equation (Eqs. 4-7 in the manuscript). This method allows for the extraction of J_0 from the slope of the linear fit to the log-transformed nucleation rate versus supersaturation data.

Correction of Data Error: We acknowledge an oversight in the conversion of nuclei count data in Supplementary Fig. 4C, which was constrained by the field of view of the microscope. We have corrected this error in our revised manuscript and confirmed that the corrected data still support our conclusions without affecting the overall trends observed.

Concerns Regarding Data Points: We recognize your concern about the potential fluctuation in the fitting results due to the limited number of saturation indices. Although this number is limited, it is consistent with the typical range used in similar studies and has proven sufficient to demonstrate clear trends in nucleation

behavior across different substrates. To further substantiate our findings, we have conducted repeated experiments (Fig. R2), which confirm the reproducibility and reliability of our results. These additional data corroborate our reported trends and enhance the robustness of our study.

Fig. R2 Repeat of the number of gypsum crystallites on the surfaces terminated with (A) $-\text{OH}$, (B) $-\text{COOH}$, (C) $-\text{NH}_2$, (D) $-\text{CH}_3$, (E) $-\text{SO}_3$ and (F) hybrid of NH_2 and COOH groups increases linearly with time at early stage of each experiment. The slopes of the plotted lines for each concentration quantify steady-state nucleation rates.

Literature Comparison: Similar or lower ionic concentrations have been reported in related literatures (Giuffre et al., 2013a; Jun et al., 2016; Li et al., 2014; Yin et al., 2022).

Reviewer 2

The study utilizes in situ microscopy to examine in real-time nucleation of gypsum on surfaces terminated with various functional groups, alongside molecular dynamics simulations. It reveals that the rate of gypsum formation is impacted by the surface functional groups and hydrophobicity. Specifically, hydrophilic surfaces exhibit surface-induced nucleation, while hydrophobic surfaces undergo bulk nucleation. This work addresses an important topic and demonstrates how different surface chemistries influence gypsum nucleation behaviour, aligning with previous studies on other mineral systems. However, the study possesses several critical flaws that currently preclude its publication in a specialised journal. Here are the main issues, listed in no particular order:

We sincerely thank the reviewer for thoroughly examining our manuscript and providing very helpful comments to guide our revision.

- The range of supersaturations employed (0.97-1.63) in the nucleation experiments is typically indicative of homogeneous nucleation, resulting in abundant nucleation within the solution bulk. Therefore, it raises questions regarding whether the authors truly observed heterogeneous nucleation, or if they instead observed nuclei forming within the bulk of the solution, which subsequently settled on the surface post-nucleation (and growth). Additionally, the absence of errors in the values of B and $\ln(A)$ complicates the assessment of whether the minor differences in these values are statistically significant or not.

We thank the reviewer for the insightful comments.

Supersaturation Range and Nucleation Type: The supersaturation range employed (0.97-1.63) in this work, calculated in natural log, was specifically chosen to facilitate the observable nucleation under optical microscopy conditions. We acknowledge that this range is typically indicative of homogeneous nucleation from the perspective of thermodynamics. However, homogeneous nucleation involved kinetics and needs several hours or days to form critical nuclei for observation (Stawski et al., 2019; Stawski et al., 2016). In our experimental system, the solutions of CaCl_2 and Na_2SO_4 are in separate reservoirs and only become supersaturated after mixing. Notably, the supersaturated solution after mixing has a very short hydraulic residence time (orders of a few seconds) before contacting the surface due to the flow rate (30 mL/h). Therefore, kinetically, homogeneous nucleation does not have the opportunity to occur. Additionally, the presence of foreign surfaces in our experiments substantially lowers the nucleation barrier, promoting heterogeneous nucleation. This is supported by previous studies (De Yoreo et al., 2013; Jun et al., 2016), which suggest that the interfacial energy between a crystal nucleus and a solid substrate is lower than that of the crystal in contact with the solution. This reduction in interfacial energy, due to stronger bonding at the substrate, significantly enhances the likelihood of heterogeneous nucleation on surfaces with functional groups.

Direct Observations of Nucleation Events: To directly observe the

nucleation process, we utilized *in situ* microscopy to monitor the initial nucleation events. This method allowed us to clearly distinguish between nuclei forming directly on the surface and those possibly settling from the bulk. Additionally, the flow rate was set at 30 mL/h to prevent the bulk nuclei from settling onto the surfaces. Moreover, we placed a COOH-functionalized substrate into a beaker with a supersaturation index of 1.42. After 30 min, the substrate was extracted from the solution and we observed the crystallite using confocal bright field microscopy. As shown in Fig. R3, the crystallites aggregated in solution and then deposited on the substrate without any specific morphology. However, in our initial experimental design, the crystallites on the substrate remained distinctly separated, each maintaining a specific morphology under steady nucleation conditions (Supplementary Fig. 3A).

Fig. R3 Confocal bright field microscopy image ($108 \times 68 \mu\text{m}$) collected from gypsum deposited on a $-\text{COOH}$ functionalized surface after nucleation in bulk solution.

Statistical Significance of Data: As for the reviewer's concern about the absence of error values for B and $\ln(A)$ and the potential for statistical fluctuations, we have conducted repeated experiments to ensure the robustness of our findings (Fig. R2). These repetitions have consistently shown similar trends in nucleation rates across different substrates, supporting the reliability and statistical significance of our reported values. The nucleation data presented in Fig. 2 and Supplementary Fig. 4, characterized by a linear dependency on time, further substantiate the independence and reproducibility of the nucleation events observed.

Fig. R2 Repeat of the number of gypsum crystallites on the surfaces terminated with (A) $-\text{OH}$, (B) $-\text{COOH}$, (C) $-\text{NH}_2$, (D) $-\text{CH}_3$, (E) $-\text{SO}_3$ and (F) hybrid of NH_2 and COOH groups increases linearly with time at early stage of each experiment. The slopes of the plotted lines for each concentration quantify steady-state nucleation rates.

Additional Evidence from Confocal Raman Spectroscopy: Confocal Raman spectroscopy was applied for post-experimentation to analyze the orientation of crystals relative to the substrates. Results demonstrate substrate-specific orientations, corroborating our assertion that nucleation predominantly occurred on the substrates, rather than in the bulk solution.

- Are the nucleation rates determined with optical microscopy comparable to the results obtained in the MD simulations? The authors utilised a 10x objective, resulting in a resolution limit of around $1.3 \mu\text{m}$, which is significantly larger than the size of nuclei. Consequently, by the time gypsum crystals were detected, they had likely already undergone substantial growth.

We acknowledge that, by the time gypsum crystals were detected, they had likely already undergone substantial growth. As mentioned in the manuscript (Lines 401-406): Gypsum nucleation on the functionalized substrate surface was monitored by optical microscopy (10 \times objective, Zeiss Co., Germany). We note that the resolution of the optical microscope was insufficient to directly observe the formation of gypsum crystallites on the substrate surface (Giuffrè et al., 2013b; Hamm et al., 2014). As a result, the nucleation rates we reported in this study were based on the assumption that each crystal observed from the optical microscope was initiated from a single crystallite and all crystal were formed heterogeneously on the substrate surface. We estimated nucleation rates by making the following two assumptions:

Each optically observed crystal was developed from a single nucleus. To say, we assumed that each nucleus, which successfully crossed the energy barrier to achieve critical size, grew into an optically observable crystallite. This assumption is reasonable because the data were collected only within the time interval where

crystallite spacing was very large (Supplementary Fig. 3A). Also, during the period of constant nucleation, each crystal consisted of a single, well-faceted rhombohedron. Moreover, the data in Fig. 2, Supplementary Fig. 4, and Fig. R2 show that their appearance was linearly dependent on time, which would not be true if nucleation events were not independent of one another.

For clarification, we have supplemented more details into the Method section of the revised manuscript as follows:

“Each crystal observed optically originated from a single nucleus. We assume that each nucleus overcame the energy barrier necessary to reach a critical size and subsequently grew into an optically observable crystallite. This assumption is substantiated by the data collected during intervals when the spacing between crystallites was significantly large (Supplementary Fig. 3A). During these intervals of steady nucleation, each crystal manifested as a distinct, well-faceted rhombohedron. Moreover, the linear relationship of crystal appearance over time, as illustrated in Fig. 2 and Supplementary Fig. 4, supports the independence of nucleation events. The absence of such linearity would suggest interdependencies among nucleation events, which was not observed.” (Lines 406-415)

As for the MD simulations, the solution concentration has a significant effect on the nucleation and crystallization of CaSO₄. A higher concentration of CaSO₄ in MD simulation is to increase the probability of ion collision and speed the simulation process. Under normal supersaturated conditions, the evolution of CaSO₄ cluster can be much slower. Additionally, we have extended the simulation time for the -CH₃ surface by an additional 5 ns and observed no significant changes in the Ca-So bond numbers (Fig. R1a), indicating that the aggregation between Ca²⁺ and SO₄²⁻ has stabilized. The time evolution of cluster number and free ions in Fig. R1(b) can also prove that the cluster aggregation has been at a stable stage.

Fig. R1 The time evolution of (a) Ca-So bond number and (b) cluster number and free ions on -CH₃ surface when the simulation time extends by 5 ns.

- The authors, in their discussion of the results, largely overlook the extensive body of literature indicating that gypsum formation follows a multistep nucleation pathway. They assert that classical nucleation is the primary mechanism behind gypsum formation, yet this assertion lacks conclusive evidence.

We would like to clarify the focus of our study and its contributions to the existing knowledge in this area as below:

Literature Review and Study Focus: While the multistep nucleation pathway has been well-documented in various studies (Demichelis et al., 2011; Du et al., 2024; Loh et al., 2017; Saha et al., 2012; Stawski et al., 2016; Van Driessche et al., 2012; Wang et al., 2012; Wang and Meldrum, 2012; Zhu et al., 2021), these have predominantly focused on homogeneous nucleation processes in supersaturated solutions. These studies typically employ advanced techniques such as high-resolution Transmission Electron Microscopy (TEM), cryo-TEM, Scanning Electron Microscopy (SEM), Raman spectroscopy, and X-ray scattering analysis, which, while informative, may introduce artifacts due to ex situ conditions and high-energy beam exposures.

Heterogeneous Nucleation Focus: In contrast, our work concentrates on gypsum nucleation occurring heterogeneously on foreign surfaces. We employed *in situ* microscopic observations to directly observe gypsum formation under fully hydrated conditions on these surfaces. This approach allows us to observe the well-ordered water molecules around functionalized surfaces, which play a crucial role in influencing the thermodynamics of nucleation and growth, as supported by previous studies (Jun et al., 2016; Navrotsky et al., 2008). Such a focus on heterogeneous nucleation provides a different perspective from the conditions studied in the aforementioned literature.

Classical vs. Multistep Nucleation in Our Study: Our findings suggest a classical nucleation pathway under the conditions studied, particularly due to the continuous hydration on functionalized substrates. This is substantiated by analyzing the relationship between nucleation rate and supersaturation index. Further corroborated by molecular dynamics (MD) simulations, clusters were observed as early as 5 ns into the simulation, with stable growth observed thereafter, as detailed in Fig. 3A and Supplementary Fig. 7. These clusters subsequently evolved into larger aggregates over time (Fig. R4), accompanied by a notable decrease in the number of free ions (Fig. 3A). Notably, the sizes of the three largest clusters underwent significant expansion in proximity to 5 ns (Fig. R5). Therefore, the inflection point at about 5 ns could be indicative of a critical point, i.e., indicative of the critical size of nuclei. These findings are consistent with the classical nucleation theory, which suggests the existence of a critical size that categorizes clusters as either subcritical or supercritical. Thus, under our experimental conditions, gypsum nucleation may not follow the multistep pathways commonly reported in other studies.

Fig. R4 The dynamic evolution of CaSO₄ cluster with time on -CH₃ surface. The clusters circled by red, green and blue are the top three largest pre-nucleation CaSO₄ clusters. Only the top 10 largest CaSO₄ clusters are displayed while the rest are hidden.

Fig. R5 The time evolution of cluster number and top three largest pre-nucleation CaSO₄ clusters on -CH₃ surface.

For clarification, we have added the two figures into the revised Supplementary Information and related discussion into the revised manuscript.

“These clusters subsequently evolve into larger aggregates over time (Supplementary Fig. 5) accompanied by a notable decrease in the number of free ions (Fig. 3A). Notably, the sizes of the three largest clusters undergo significant expansion in proximity to 5 ns (Supplementary Fig. 6). Therefore, the inflection point at about 5 ns could be indicative of a critical point, indicative of the critical size of nuclei. These findings are consistent with the classical nucleation theory, which suggests the existence of a critical size that categorizes clusters as either subcritical or supercritical.” (Lines 157-163)

- The simulations were conducted at a concentration of approximately 32 M, resulting in cluster formation under conditions significantly different from those in the experiments. Moreover, at such high concentrations, the pH level becomes a critical factor to consider. Additionally, the simulation time is notably brief. Ideally, it should be increased by one order of magnitude.

We have the following response to address the reviewer 's concern:

Concentration Justification: The concentration of CaSO₄ in our simulations was set at 0.463 M, which is indeed higher than the experimental conditions (0.05, 0.07, 0.085, 0.1 M). This selection was driven by the inherent limitations of MD simulations related to time and spatial scales. Lower concentrations, more akin to experimental conditions, would require significantly longer simulation times to observe pre-nucleation cluster formation due to the decreased probability of ion collisions. The chosen concentration is a compromise that increases the likelihood of observing significant interactions within feasible computational times and is a practice supported by precedent in the field (Li et al., 2021; Smeets et al., 2017).

pH Considerations: Regarding the pH levels at high concentrations, we acknowledge that this is an important factor in real-world scenarios. However, simulating the effects of pH in MD simulations presents substantial challenges, particularly in terms of accurately modeling the ionization states of functional groups under varying pH conditions. In our simulations, we assumed a fully ionized state for the functional groups on the surfaces (-Hybrid, -COOH, -NH₂, and SO₃-) to maintain consistency and computational feasibility.

Simulation Time: The simulation duration of 20 ns, while relatively short in a broader context, represents a significant computational effort and is consistent with standard practices within the field for similar studies. Extending the simulation time by an order of magnitude would indeed provide more extensive data, but at a prohibitive computational cost. Importantly, the trends we observed, such as the Ca-So bond number stabilization shown in Fig. R1, indicate that the major nucleation events had reached a relatively stable state within the 20 ns timeframe. This observation supports our decision to limit the simulation time to balance detail with feasibility.

- Conducting multiple simulations and performing statistical analysis are crucial in Molecular Dynamics (MD) studies, particularly those focusing on aggregation. The initial distribution of molecules significantly impacts the formation of aggregates of different sizes.

We appreciate the reviewer's insightful observation regarding the significance of conducting multiple simulations and the necessity for rigorous statistical analysis in MD studies, especially those that examine aggregation processes. We concur that the initial distribution of molecules indeed influences the formation and subsequent evolution of CaSO₄ clusters. In response to the reviewer's concern, our MD simulations are structured to include both a relaxation phase and an equilibrium phase. During the relaxation phase, although the distribution of molecules varies and evolves, these configurations are not utilized for the final data

analysis to avoid biases introduced by non-equilibrium states.

To ensure robust and reliable results, our analyses are based on the data collected during the equilibrium phase of the simulations. Specifically, for this study, the total simulation duration was set at 20 ns, with statistical analyses performed on averaged data from the last 5 ns of each run. This approach helps mitigate any anomalies that may arise from specific initial distributions, as averaging over this period allows for the observation of more stable, representative aggregation behaviors.

Other comments:

The characterisation of the functionalised surfaces, prepared following standard protocols, should be included in the Materials and Methods section or supplementary information, rather than in the Results section.

In this study, the relationship between gypsum heterogeneous nucleation pathways and surface functional groups and hydrophobicity was investigated. The hydrophilicity and functionality of the surface are critical data, while the roughness of surface is marginal data. Thus, we put the roughness data in the Supplementary Information, while keep the data of hydrophilicity and functionality in the Result section.

The volume of the solution in the MD simulations is not clear because the authors did not mention the amount of space occupied by the substrate in the simulation box.

The dimension of the simulation box is $5.77 \text{ nm} \times 5.77 \text{ nm} \times 14.56 \text{ nm}$, and the solution is consisting of 9000 water molecules and 75 pairs of ions, thus, the volume of the aqueous solution is about $2.7 \times 10^5 \text{ nm}^3$. The solution volume has been also added in the revised manuscript.

“and the volume of the aqueous solution is about $2.7 \times 10^5 \text{ nm}^3$.” (Lines 438-439)

Taking the $-\text{CH}_3$ surface as an example, during the initial phase, the rapid decay of free ions coincides with the increase of precursor clusters (Fig. 3A), which manifest as chains, branches, and rings, consistent with prior studies on calcium carbonate nucleation^{3, 26}. In the subsequent phase, these precursor clusters further amalgamate into larger, amorphous structures.

>> These findings appear to closely align with what has been reported for CaSO_4 nucleation in the literature. Therefore, it would be more appropriate to discuss those works instead of focusing on CaCO_3 .

Accepting the reviewer’s suggestion, we have discussed the following references in the revised manuscript:

R1. Stawski, T. M. et al. The Structure of CaSO_4 Nanorods: The Precursor of Gypsum. *J. Phys. Chem. C* 123, 23151–23158 (2019).

R2. Li, H.-J. et al. Structures and dynamic hydration of CaSO_4 clusters in supersaturated solutions: A molecular dynamics simulation study. *Journal of Molecular Liquids* 324, 115104 (2021).

“Taking the $-CH_3$ surface as an example, during the initial phase, the rapid decay of free ions coincides with the increase of precursor clusters (Fig. 3A), which manifest as chains, branches, and rings, consistent with prior studies on calcium sulfate nucleation.” (Lines 154-157)

>> Additionally, this pathway does not conform to classical nucleation theory (CNT). Therefore, it is unclear why the authors assert that heterogeneous gypsum nucleation is classical. For instance, in Line 294: Experimental and simulation results corroborate that gypsum nucleation aligns with classical nucleation theory.

As we respond above, our findings suggest a classical nucleation pathway under the conditions studied, particularly due to the continuous hydration on functionalized substrates. This was substantiated by analyzing the relationship between the nucleation rate and supersaturation index (Fig. 2). Further corroborated by molecular dynamics (MD) simulations, clusters were observed as early as 5 ns into the simulation, with stable growth observed thereafter, as detailed in Fig. 3A and Supplementary Fig. 7. These clusters subsequently evolved into larger aggregates over time (Fig. R4), accompanied by a notable decrease in the number of free ions (Fig. 3A). Notably, the sizes of the three largest clusters underwent significant expansion in proximity to 5 ns (Fig. R5). Therefore, the inflection point at about 5 ns could be indicative of a critical point, i.e., indicative of the critical size of nuclei. These findings are consistent with the classical nucleation theory, which suggests the existence of a critical size that categorizes clusters as either subcritical or supercritical. Thus, under our experimental conditions, gypsum nucleation follows the classical nucleation theory.

As a result, the nucleation rates we reported in this study were based on the assumption that each crystal observed from the optical microscope was initiated from a single crystallite and all crystals were formed heterogeneously on the substrate surface.

>> What evidence supports this assumption?

Each optically observed crystal was developed from a single nucleus. That is, we assumed that each nucleus, which successfully crossed the energy barrier to achieve critical size, grew into an optically observable crystallite. The confocal Raman result (Supplementary Fig. 4A) could support this assumption. The data were collected only within the time interval where crystallite spacing was very large, which would be aggregates if the crystallites formed homogeneously on the substrate surface (Fig. R2). Moreover, the data in Fig. 2 and Supplementary Fig. 4 show that their appearance was linearly dependent on time, which would not be true if nucleation events were not independent of one another.

Reviewer 3

The manuscript concerns the nucleation of gypsum on functionalised SAMs functionalised surfaces. It allowed the authors to control the wettability of substrates from hydrophobic to hydrophilic. The authors performed growth experiments involving a flow setup and optical microscopy. Further insights were gained with molecular dynamics simulations.

In overall, I find the idea of using SAMs to affect gypsum nucleation relatively novel and interesting. However, the combination of used methods is insufficient to support the conclusions of the paper. I have recognised the following weaknesses:

We are grateful to the reviewer for his/her time and effort in reviewing our manuscript. Our response to the comments is detailed below with revisions to the manuscript highlighted.

1. The authors focus extensively on the nanometre length-scale aspects of differences in nucleation on hydrophobic and hydrophilic surfaces, but this is based only MD simulations. On the other hand, their experiments probe the length-scale of many microns, which is accessible to optical microscopy. In fact, they even state this in the SI: "As a result, the nucleation rates we reported in this study were based on the assumption that each crystal observed from the optical microscope was initiated from a single crystallite and all crystal were formed heterogeneously on the substrate surface."

In this regard, optical microscopy is not suitable to study nucleation. It is useful merely for further stages of crystallisation: the nuclei (or more broadly precursors) of CaSO₄ are maybe a few nm in size, whereas optical microscopy in practice can consider objects of hundreds on nm upwards. Nucleation is not crystallisation, and I find the experimental results quite unuseful to say anything about nucleation. If anything, I would focus only on the MD results.

We recognize the inherent limitations of optical microscopy in directly observing nucleation events, which typically occur at the nanometer scale. However, our MD simulations aim to bridge this gap by providing detailed insights into the early stages of nucleation, which we infer to influence the larger scale crystallization processes observed.

The use of optical microscopy in our study was intended primarily to observe the outcomes of nucleation, i.e., the growth stages of crystals that follow nucleation. These observations were complemented by MD simulations, which provided insights into the initial nucleation events at the molecular level. Although we recognize the limitations of optical microscopy for direct nucleation observation, we used it to indirectly infer nucleation rates and mechanisms based on the growth and distribution of crystallites observable at the micron scale.

In our experimental design, we ensured that each crystal observed via optical microscopy was originated from a single nucleation event, supported by the growth dynamics reported (Fig. 2 and Supplementary Fig. 4). Their appearance was

linearly dependent on time, which would not be true if nucleation events were not independent of one another. To further validate these observations, we employed confocal Raman spectroscopy, which corroborates the assumption that observed crystals developed independently from single nucleation sites (Supplementary Fig. 3A), in consistent with previous studies (Giuffre et al., 2013a). Moreover, we placed a COOH-functionalized substrate into a beaker with a supersaturation index of 1.42. After 30 min, the substrate was extracted from the solution and we observed the crystallite using confocal bright field microscopy. As shown in Fig. R3, the crystallites would aggregate and deposit on the substrate with no specific morphology. However, the crystallites heterogeneously nucleated on the substrate separated from each other with specific morphology (Supplementary Fig. 3A).

Fig. R3 Confocal bright field microscopy image ($108 \times 68 \mu\text{m}$) collected from the gypsum deposited on a $-\text{COOH}$ functionalized surface after nucleation in bulk solution.

2. The very fact that a process is describable by the classical equations (of nucleation and crystallisation), does not mean that it is classical. This is well-explained by Gebauer and Wolf in JACS:

<https://pubs.acs.org/doi/10.1021/jacs.8b13231>

In this regard, classical NT is a mathematical framework which often works well, but it does not explain the mechanism (or all its microscopi details). It ASSUMES a certain mechanism, which often fits the observation for some length scale. On the other, the non-classical NT, currently lacks such a robust framework, and usually recognises the fact that nucleation is not driven by solute ions, but larger entities. Consequently, the observation of such entites is crucial to say that the processes is non-classical. Moreover, as is highlighted in the referenced article, it appears that the CNT is simply a model, while in reality all (most) mineral nucleation processes should be considered non-classical.

Thank you for the insightful comments and for directing us to the similar work by Gebauer and Wolf, which underscores the complexity of nucleation beyond classical descriptions. We acknowledge the limitations of Classical Nucleation Theory (CNT), which, while providing a valuable framework for understanding nucleation under certain conditions, might not capture all microscopic details of complex systems like gypsum formation.

In our study, CNT is employed because of its well-established predictive value, particularly in settings where the classical theory has been validated. However, recognizing the potential shortcomings of CNT in explaining every aspect of nucleation, we have also integrated Molecular Dynamics (MD) simulations to probe the atomic-level interactions and dynamics that contribute to nucleation. These simulations reveal the formation of clusters as early as 5 ns (Fig. 3A and Supplementary Fig. 7). These clusters subsequently evolved into larger aggregates over time (Fig. R4), accompanied by a notable decrease in the number of free ions (Fig. 3A). Notably, the sizes of the three largest clusters underwent significant expansion in proximity to 5 ns (Fig. R5). Therefore, the inflection point at about 5 ns could be indicative of a critical point, i.e., indicative of the critical size of nuclei. These findings are consistent with the classical nucleation theory, which suggests the existence of a critical size that categorizes clusters as either subcritical or supercritical (Sleutel et al., 2014; Van Driessche et al., 2021). Thus, under our experimental conditions, gypsum nucleation follows the classical nucleation pathways.

Fig. R4 The dynamic evolution of CaSO_4 cluster with time on $-\text{CH}_3$ surface. The clusters circled by red, green and blue are the top three largest pre-nucleation CaSO_4 clusters. Only the top 10 largest CaSO_4 clusters are displayed while the rest are hidden.

Fig. R5 The time evolution of cluster number and top three largest pre-nucleation CaSO_4 clusters on $-\text{CH}_3$ surface.

For clarification, we have added the two figures into the revised Supplementary Information and related discussion in the revised manuscript.

“These clusters subsequently evolve into larger aggregates over time (Supplementary Fig. 5) accompanied by a notable decrease in the number of free ions (Fig. 3A). Notably, the sizes of the three largest clusters undergo significant expansion in proximity to 5 ns (Supplementary Fig. 6). Therefore, the inflection point at about 5 ns could be indicative of a critical point, indicative of the critical size of nuclei. These findings are consistent with the classical nucleation theory, which suggests the existence of a critical size that categorizes clusters as either subcritical or supercritical.” (Lines 157-163)

Furthermore, we have also searched a wide range of literatures about non-classical nucleation theory (Demichelis et al., 2011; Du et al., 2024; Loh et al., 2017; Saha et al., 2012; Stawski et al., 2016; Van Driessche et al., 2012; Wang et al., 2012; Wang and Meldrum, 2012; Zhu et al., 2021). These studies have predominantly focused on homogeneous nucleation processes in supersaturated solutions. These studies typically employ advanced techniques such as high-resolution Transmission Electron Microscopy (TEM), cryo-TEM, Scanning Electron Microscopy (SEM), Raman spectroscopy, and X-ray scattering analysis, which, while informative, may introduce artifacts due to *ex situ* conditions and high-energy beam exposures. In contrast, our work concentrates on gypsum nucleation occurring heterogeneously on foreign surfaces. We employed *in situ* microscopic observations to directly investigate gypsum formation under fully hydrated conditions on these surfaces. This approach allows us to observe the well-ordered water molecules around functionalized surfaces, which play a crucial role in influencing the thermodynamics of nucleation and growth, as supported by previous studies (Jun et al., 2016; Navrotsky et al., 2008). Such a focus on heterogeneous nucleation provides a different perspective from the conditions studied in the aforementioned literature.

In summary, while we utilize CNT as a foundational framework, we critically

expand it with detailed molecular insights from MD simulations and empirical observations from advanced microscopy techniques, ensuring a robust analysis of gypsum nucleation on functionalized surfaces.

To address the reviewer's concerns, we have added more details about the mechanism in the Discussion section in the revised manuscript:

“The non-classical nucleation model is arguably the most popular on gypsum nucleation. Direct experimental evidence for a gypsum nucleation starts from the aggregation of sub-3 nm primary species or CaSO₄ precursor, revealing that the nucleation of gypsum occurs nonclassically (Stawski et al., 2019; Stawski et al., 2016; Van Driessche et al., 2012). These have predominantly focused on homogeneous nucleation processes in supersaturated solutions. These studies typically employ advanced techniques such as high-resolution Transmission Electron Microscopy (TEM), cryo-TEM, Scanning Electron Microscopy (SEM), Raman spectroscopy, and X-ray scattering analysis, which, while informative, may introduce artifacts due to ex situ conditions and high-energy beam exposures (Du et al., 2024; Stawski et al., 2019; Stawski et al., 2016; Van Driessche et al., 2012).

In contrast, our research concentrates on gypsum nucleation occurring heterogeneously on foreign surfaces. We employed in situ microscopic observations to directly investigate gypsum formation under fully hydrated conditions on these surfaces. This approach allows us to observe the well-ordered water molecules around functionalized surfaces, which play a crucial role in influencing the thermodynamics of nucleation and growth, as supported by previous studies (Jun et al., 2016; Navrotsky et al., 2008). This focus on heterogeneous nucleation provides a different perspective from the conditions studied in the aforementioned literature.” (Lines 297-312)

3. Either way, this can be established only based on nano-scale observations. The authors, indicate the presence of cluster entities in their MD, which I would interpret as a clear sign of the processes being non-classical. The fact that classical equations also work here is in a sense irrelevant. This has been observed for many mineral systems, most notably CaCO₃, which we know that exhibits all sorts of non-classical features: PNCs, amorphous phases, mesocrystallinity etc.

We would like to clarify the focus of our study and its contributions to the existing knowledge in this area as below:

Relevance to Existing Literature: While the non-classical nucleation pathway has been well-documented in various studies (Demichelis et al., 2011; Du et al., 2024; Loh et al., 2017; Saha et al., 2012; Stawski et al., 2016; Van Driessche et al., 2012; Wang et al., 2012; Wang and Meldrum, 2012; Zhu et al., 2021), these have predominantly focused on homogeneous nucleation processes in supersaturated solutions. These studies typically employ advanced techniques such as high-resolution TEM, cryo-TEM, SEM, Raman spectroscopy, and X-ray scattering analysis, which, while informative, may introduce artifacts due to ex situ conditions and high-energy beam exposures.

Focus of Our Work: In contrast, our work concentrates on gypsum

nucleation occurring heterogeneously on foreign surfaces. We employed *in situ* microscopic observations to directly investigate gypsum formation under fully hydrated conditions on these surfaces. This approach allows us to observe the well-ordered water molecules around functionalized surfaces, which play a crucial role in influencing the thermodynamics of nucleation and growth, as supported by previous studies (Jun et al., 2016; Navrotsky et al., 2008). Such a focus on heterogeneous nucleation provides a different perspective from the conditions studied in the aforementioned literature.

Classical Nucleation in Our Work: Our findings suggest a classical nucleation pathway under the conditions studied, particularly due to the continuous hydration on functionalized substrates. This conclusion was substantiated by analyzing the relationship between nucleation rate and supersaturation index. Further corroborated by molecular dynamics simulations, as we responded above.

4. In fact it has been found in several studies recently, that CaSO₄ exhibits a mesostructure in its single crystals, which seems to be linked the particle-mediated crystallisation. These earlier observations, at a first glance, maybe be linked to clusters in MD, as observed by the authors.

The particle-mediated growth has already been mentioned in ref. 15, but has been further elaborated by Stawski et al. in:

<https://doi.org/10.1021/acs.cgd.9b00066>

<https://www.pnas.org/doi/10.1073/pnas.2111213118>

We understand the reviewer's concern about the mesostructure in gypsum single crystals. Most mesocrystals are formed via oriented aggregation and transit to single crystal. During the formation process, templating (macro)molecules are identified so far to control the particle arrangement in a highly ordered (Shen et al., 2013; Van Driessche et al., 2021; Yuwono et al., 2010; Zhu et al., 2021).

We further analyzed the clusters in MD and demonstrated that they were critical nuclei, which subsequently evolved into larger aggregates over time (Figs. R4 and R5). This observation aligns with classical nucleation theory, wherein a critical size must be reached before stable growth can proceed. Additionally, we focused on heterogeneous nucleation on a foreign surface under hydration conditions. This approach is distinct from the homogeneous nucleation, which is often associated with mesostructural formation, thereby providing a unique perspective on the nucleation process.

5. It should be clearly distinguished what constitutes non-classical nucleation and what non-classical crystallisation. The non-classical nucleation assumes the involvement of cluster species i.e. PNCs, while crystallisation is non-classical when it is multi-stage with transitional phases. Furthermore, we can also consider the subsequent growth stage. In this regard, for instance, optical microscopy results consider further stages of crystallisation and maybe growth, and the classical framework is used to back-extrapolate the results to the nucleation stage. This does not prove that nucleation is non-classical. Moreover, as is stated by D. Gebauer, it appears that many mineral

systems nucleate non-classically, while crystallisation seems to follow a classical view. These are not contradictory!

We fully recognize the importance of differentiating between non-classical nucleation, which often involves the formation of pre-nucleation clusters (PNCs), and non-classical crystallization, characterized by multi-stage transitions and the growth of these clusters into mature crystals. These distinctions are crucial for accurately describing the processes observed in our study.

To address the reviewer's concern, we have taken great care to use molecular dynamics (MD) simulations to complement our optical microscopy observations, specifically the initial stages of nucleation. The MD simulations are designed to capture the dynamics at a molecular level, allowing us to observe the formation and behavior of clusters at stages too early to be detected by optical microscopy. Our MD simulations show that the initial nucleation process can be described well within a classical framework, where a critical nucleus forms and grows. This aligns with the patterns observed at later stages via optical microscopy, where the growth of the nuclei is visible.

6. More effort should be put into referencing current (and less current) literature of CaSO₄ nucleation and crystallisation, and also other mineral systems.

Accepting the reviewer suggestion, we have replaced some references with more up-to-date ones to better reflect the latest trend of the relevant studies in the revised manuscript.

60. Du, J.S., Bae, Y. & De Yoreo, J.J. Non-classical crystallization in soft and organic materials. *Nature Reviews Materials* **9**, 229-248 (2024).

63. Li, C., Liu, Z., Goonetilleke, E.C. & Huang, X. Temperature-dependent kinetic pathways of heterogeneous ice nucleation competing between classical and non-classical nucleation. *Nature Communications* **12**, 4954 (2021).

64. Van Driessche, A.E.S. et al. Molecular nucleation mechanisms and control strategies for crystal polymorph selection. *Nature* **556**, 89-94 (2018).

65. Bai, G., Gao, D., Liu, Z., Zhou, X. & Wang, J. Probing the critical nucleus size for ice formation with graphene oxide nanosheets. *Nature* **576**, 437-441 (2019).

References

- De Yoreo, J.J., Waychunas, G.A., Jun, Y.-S. and Fernandez-Martinez, A. 2013. In situ Investigations of Carbonate Nucleation on Mineral and Organic Surfaces. *Reviews in Mineralogy and Geochemistry* **77**(1), 229-257.
- Demichelis, R., Raiteri, P., Gale, J.D., Quigley, D. and Gebauer, D. 2011. Stable prenucleation mineral clusters are liquid-like ionic polymers. *Nature Communications* **2**(1), 590.
- Du, J.S., Bae, Y. and De Yoreo, J.J. 2024. Non-classical crystallization in soft and organic materials. *Nature Reviews Materials* **9**(4), 229-248.
- Giuffre, A.J., Hamm, L.M., Han, N., De Yoreo, J.J. and Dove, P.M. 2013a. Polysaccharide chemistry regulates kinetics of calcite nucleation through

- competition of interfacial energies. *Proceedings of the National Academy of Sciences* 110(23), 9261-9266.
- Giuffre, A.J., Hamm, L.M., Han, N., De Yoreo, J.J. and Dove, P.M. 2013b. Polysaccharide chemistry regulates kinetics of calcite nucleation through competition of interfacial energies. *Proc. Natl. Acad. Sci. USA* 110(23), 9261.
- Hamm, L.M., Giuffre, A.J., Han, N., Tao, J., Wang, D., De Yoreo, J.J. and Dove, P.M. 2014. Reconciling disparate views of template-directed nucleation through measurement of calcite nucleation kinetics and binding energies. *Proc. Natl. Acad. Sci. USA* 111(4), 1304.
- Jun, Y.-S., Kim, D. and Neil, C.W. 2016. Heterogeneous Nucleation and Growth of Nanoparticles at Environmental Interfaces. *Accounts of Chemical Research* 49(9), 1681-1690.
- Li, H.-J., Wang, C.-C., Wang, M., Zhang, Q.-W., Li, Y.-Y., Yi, H.-B. and Chen, Y. 2021. Structures and dynamic hydration of CaSO₄ clusters in supersaturated solutions: A molecular dynamics simulation study. *Journal of Molecular Liquids* 324, 115104.
- Li, Q., Fernandez-Martinez, A., Lee, B., Waychunas, G.A. and Jun, Y.-S. 2014. Interfacial Energies for Heterogeneous Nucleation of Calcium Carbonate on Mica and Quartz. *Environmental Science & Technology* 48(10), 5745-5753.
- Loh, N.D., Sen, S., Bosman, M., Tan, S.F., Zhong, J., Nijhuis, C.A., Král, P., Matsudaira, P. and Mirsaidov, U. 2017. Multistep nucleation of nanocrystals in aqueous solution. *Nature Chemistry* 9(1), 77-82.
- Navrotsky, A., Mazeina, L. and Majzlan, J. 2008. Size-Driven Structural and Thermodynamic Complexity in Iron Oxides. *Science* 319(5870), 1635-1638.
- Saha, A., Lee, J., Pancera, S.M., Bräeu, M.F., Kempter, A., Tripathi, A. and Bose, A. 2012. New insights into the transformation of calcium sulfate hemihydrate to gypsum using time-resolved cryogenic transmission electron microscopy. *Langmuir : the ACS journal of surfaces and colloids* 28 30, 11182-11187.
- Shen, J.-W., Li, C., van der Vegt, N.F.A. and Peter, C. 2013. Understanding the Control of Mineralization by Polyelectrolyte Additives: Simulation of Preferential Binding to Calcite Surfaces. *The Journal of Physical Chemistry C* 117(13), 6904-6913.
- Sleutel, M., Lutsko, J., Van Driessche, A.E.S., Durán-Olivencia, M.A. and Maes, D. 2014. Observing classical nucleation theory at work by monitoring phase transitions with molecular precision. *Nature Communications* 5(1), 5598.
- Smeets, P.J.M., Finney, A.R., Habraken, W.J.E.M., Nudelman, F., Friedrich, H., Laven, J., De Yoreo, J.J., Rodger, P.M. and Sommerdijk, N.A.J.M. 2017. A classical view on nonclassical nucleation. *Proceedings of the National Academy of Sciences* 114(38), E7882-E7890.
- Stawski, T.M., Van Driessche, A.E.S., Besselink, R., Byrne, E.H., Raiteri, P., Gale, J.D. and Benning, L.G. 2019. The Structure of CaSO₄ Nanorods: The Precursor of Gypsum. *J. Phys. Chem. C* 123(37), 23151-23158.
- Stawski, T.M., van Driessche, A.E.S., Ossorio, M., Diego Rodriguez-Blanco, J., Besselink, R. and Benning, L.G. 2016. Formation of calcium sulfate

- through the aggregation of sub-3 nanometre primary species. *Nature Communications* 7(1), 11177.
- Van Driessche, A.E.S., Benning, L.G., Rodriguez-Blanco, J.D., Ossorio, M., Bots, P. and García-Ruiz, J.M. 2012. The Role and Implications of Bassanite as a Stable Precursor Phase to Gypsum Precipitation. *Science* 336(6077), 69-72.
- Van Driessche, A.E.S., Van Gerven, N., Joosten, R.R.M., Ling, W.L., Bacia, M., Sommerdijk, N. and Sleutel, M. 2021. Nucleation of protein mesocrystals via oriented attachment. *Nature Communications* 12(1), 3902.
- Wang, Y.-W., Kim, Y.-Y., Christenson, H.K. and Meldrum, F.C. 2012. A new precipitation pathway for calcium sulfate dihydrate (gypsum) via amorphous and hemihydrate intermediates. *Chemical Communications* 48(4), 504-506.
- Wang, Y.-W. and Meldrum, F.C. 2012. Additives stabilize calcium sulfate hemihydrate (bassanite) in solution. *Journal of Materials Chemistry* 22(41), 22055-22062.
- Yin, Y., Li, T., Zuo, K., Liu, X., Lin, S., Yao, Y. and Tong, T. 2022. Which Surface Is More Scaling Resistant? A Closer Look at Nucleation Theories for Heterogeneous Gypsum Nucleation in Aqueous Solutions. *Environmental Science & Technology* 56(22), 16315-16324.
- Yuwono, V.M., Burrows, N.D., Soltis, J.A. and Penn, R.L. 2010. Oriented Aggregation: Formation and Transformation of Mesocrystal Intermediates Revealed. *Journal of the American Chemical Society* 132(7), 2163-2165.
- Zhu, G., Sushko, M.L., Loring, J.S., Legg, B.A., Song, M., Soltis, J.A., Huang, X., Rosso, K.M. and De Yoreo, J.J. 2021. Self-similar mesocrystals form via interface-driven nucleation and assembly. *Nature* 590(7846), 416-422.

Response to Reviewer 1 Comments

In the revised manuscript, SI and the reply of Guan et al, the authors have adequately addressed all my queries and I believe that their simulations and experimental results are reliable. I hold reservations regarding the author's adherence to CNT interpretation for their results. The induced effect at the interface may accelerate nucleation, probably allowing the study to be described using CNT. Whether CNT or non- CNT, clusters will appear during the nucleation processes, whom are different in stability. The presence of stable particles or structures in experiments (interrupted or delayed evolution) is a typical characteristics of non-CNT mechanisms. Observations in experiments are already crystallite instead of crystal nuclei, while clusters in simulations are more likely pre-nucleation clusters. If they can rapidly evolve and grow, they should fall within the realm of CNT. Therefore, if the authors can further substantiate their claims by examining the evolving properties or dynamic changes of clusters during their simulations, or adjust their expression according to the points of referee III on NCNT, I am inclined towards publication of this manuscript. Even after the modifications, the related viewpoints are still worth further exploration and validation, while their results would provide readers with a reference and discussion space or opportunity.

Thank you for your thorough review and insightful comments. We appreciate the opportunity to address your concerns.

1. Examining Evolving Properties or Dynamic Changes of Clusters: We extended the simulation time for the -CH₃ surface to 50 ns to analyze the evolving properties and dynamic changes of clusters. Results in **Fig. R1** and **Supplementary Fig. 5** show that these clusters gradually evolved into larger formations, with the size of the top two clusters experiencing minimal changes after 20 ns. A closer examination of the structural descriptors, specifically the radial distribution function (RDF) and coordination number within the clusters, reveals that the RDF of calcium atoms with oxygen atoms of sulfate increased sharply from 0 to 20 ns and then plateaued (**Fig. R2a**), accompanied by an increase in the coordination number from 0 to 3 (**Fig. R2c**). Conversely, the RDF of calcium atoms with oxygen atoms of water decreased from 0 to 20 ns and then plateaued (**Fig. R2b**), with the coordination number decreasing from 7 to 3 (**Fig. R2c**). These results collectively indicate that the evolution of clusters aligned with the CNT framework, growing after the critical point and eventually reaching a stable state.

Fig. R1 The dynamic evolution of CaSO_4 cluster with time on $-\text{CH}_3$ surface in the duration of 25-50 ns. Only the CaSO_4 clusters were displayed while the rest were hidden.

Fig. R2 The radial distribution function, $g(r)$, of calcium atoms with (a) oxygen atoms of sulfate (Os) and (b) oxygen atoms of water (Ow) with time. (c) The coordination number of calcium atoms with oxygen atoms of sulfate and water with time.

2. Phase of CaSO_4 : Non-classical nucleation process describes the formation of new phases via the aggregation of individual molecules or atoms. This process involves intermediate species and complex pathways that can lead to the formation of crystalline structures. To investigate whether the crystal was developed from intermediate phase of CaSO_4 , surface plasmon resonance microscopy (SPRM) was employed. This technique allows for the high-throughput identification of nuclei by accurately measuring the refractive index of individual nuclei without interference from background signals (Qian et al., 2019; Wu et al., 2024). Since the refractive index of crystalline materials is often higher than that of amorphous or liquid materials, the phase transformation of CaSO_4 in the early nucleation process could be identified by this method, which would show a sudden signal change. By tracking the trajectories of CaSO_4 formed on the surface, a time-resolved plasmonic image sequence was captured (**Fig. R3a**). Upon nucleation of CaSO_4 , a typical scattering pattern was clearly observed. As the time progressed, the plasmonic intensity increased (**Fig. R3b**), indicating CaSO_4 nuclei growth. Additionally, the

monotonically increased plasmonic intensity excluded a phase change in CaSO_4 . This result is consistent with the CNT, which suggests the existence of a critical size that categorizes clusters as either subcritical or supercritical. Thus, under our experimental conditions, intermediate phase of CaSO_4 might not exist as suggested in other studies.

Fig. R3 (a) Snapshot plasmonic images and (b) intensity tracking of CaSO_4 nuclei during the nucleation process. The snapshot when the signal begins to appear was marked as the first frame ($T = 0$ s). Conditions: The plasmonic imaging system utilized a commercial inverted microscope (Ti microscope, Nikon Co., Japan) with a $100\times$ oil-immersion objective lens ($\text{NA} = 1.49$). Illumination was provided by a 660 nm superluminescent diode to excite surface plasmon resonance. Standard glass slides were replaced with sensing chips modified by self-assembled monolayer with COOH groups. Plasmonic images were recorded by a CCD camera (Pike-032B, Allied Vision Technologies Co., USA) through a $0.46\times$ zoom-out lens (Nikon Co., Japan).

Additionally, we calculated the significant changes in the RDF of sulfur atom pairs of sulfate over time. To identify the specific structure of the clusters, the RDF diagrams of amorphous phase and perfect crystal of CaSO_4 were compared. As illustrated in **Fig. R4**, the RDF peaks of amorphous calcium sulfate were broad and flattened, while the RDF peaks of crystalline calcium sulfate were sharp and well-defined, exhibiting a degree of long-range order. The simulations performed for the clusters over the period from 0 to 50 ns show that the behavior of the clusters was similar to that of a crystal phase. Therefore, from an atomic-scale perspective, calcium ions and sulfate ions aggregated together, arranging themselves in a

crystalline form. These results collectively suggest that the intermediate phase, commonly observed in non-classical nucleation processes, could be excluded in this study.

Fig. R4 The radial distribution function, $g(r)$, of sulfur atom pairs of sulfate with time. CaSO_4 crystal and amorphous phase of CaSO_4 were calculated for comparison.

To address the reviewer’s concern, we have added the two figures (from the new experimental results) into the revised Supplementary Information and related discussion into the revised manuscript as follows:

“Given recent observations of non-classical gypsum nucleation, it is striking to see how gypsum nucleation on foreign surfaces aligns the classical framework. We conducted surface plasmon resonance microscopy experiments, which allow for the nanoscale spatiotemporal identification of nuclei by accurately measuring the refractive index of individual nuclei without interference from background signals. Since the refractive index of crystalline materials is often higher than that of amorphous or liquid materials, the phase transformation of CaSO_4 in the early nucleation process could be identified by this method, which would show a sudden signal change. By tracking the trajectories of CaSO_4 formed on the surface, a time-resolved plasmonic image sequence was captured (Supplementary Fig. 15a). Upon nucleation, a typical scattering pattern was clearly observed. As the time progressed, the plasmonic intensity increased (Supplementary Fig. 15b), indicating CaSO_4 nuclei growth. Notably, the monotonically increased plasmonic intensity excluded a phase change in CaSO_4 . Additionally, we calculated the great changes in the RDF of sulfur atom pairs of sulfate with time. To identify the specific structure of the clusters, the RDF diagrams of amorphous phases and perfect crystal of CaSO_4 were compared. As illustrated in Supplementary Fig. 16, the RDF peaks of amorphous calcium sulfate were broad and flattened, while the RDF peaks of crystalline calcium sulfate were sharp and well-defined, exhibiting a degree of long-range order. Simulations performed for the clusters over the period from 0 to 50 ns show that the behavior of the clusters was similar to that of a crystal phase. Therefore, the intermediate phase, commonly observed in non-classical nucleation process, could be excluded in this study.” (Lines 322-341)

Response to Reviewer 2 Comments

The authors have partially addressed the issues raised by the reviewer, but overall, the main problem persists: the presented results do not allow to substantiate the claim that heterogeneous nucleation follows a classical nucleation pathway or a non-classical pathway. I do not see how the authors will resolve this without performing additional experimental work to probe the nucleation events in situ with high enough resolution to be able to observe the precise nucleation mechanism(s).

Thank you for your feedback. We appreciate your thorough review and the opportunity to improve our manuscript further.

We understand your concerns regarding the nucleation pathway. Non-classical nucleation pathway often involves amorphous phase before reaching crystallite. To investigate whether there is amorphous phase at early stages of gypsum formation. We conducted in situ nanoscale spatiotemporal surface plasmon resonance microscopy (SPRM) experiments and examined the structural descriptor (radial distribution function of sulfur atom pairs of sulfate). Results reveal that gypsum nucleation on a foreign surface followed a classical pathway.

1. Surface Plasmon Resonance Microscopy Experiments: Non-classical nucleation process describes the formation of new phases via the aggregation of individual molecules or atoms. This process involves intermediate species and complex pathways that can lead to the formation of crystalline structures. To investigate whether the crystal was developed from intermediate phase of CaSO_4 , surface plasmon resonance microscopy (SPRM) was employed. This technique allows for the high-throughput identification of nuclei by accurately measuring the refractive index of individual nuclei without interference from background signals (Qian et al., 2019; Wu et al., 2024). Since the refractive index of crystalline materials is often higher than that of amorphous or liquid materials, the phase transformation of CaSO_4 in the early nucleation process could be identified by this method, which would show a sudden signal change. By tracking the trajectories of CaSO_4 formed on the surface, a time-resolved plasmonic image sequence was captured (**Fig. R3a**). Upon nucleation of CaSO_4 , a typical scattering pattern was clearly observed. As the time progressed, the plasmonic intensity increased (**Fig. R3b**), indicating CaSO_4 nuclei growth. Additionally, the monotonically increased plasmonic intensity excluded a phase change in CaSO_4 . This result is consistent with the CNT, which suggests the existence of a critical size that categorized clusters as either subcritical or supercritical. Thus, under our experimental conditions, intermediate phase of CaSO_4 might not exist as suggested in other studies.

Fig. R3 (a) Snapshot plasmonic images and (b) intensity tracking of CaSO_4 nuclei during the nucleation process. The snapshot when the signal begins to appear was marked as the first frame ($T = 0$ s). Conditions: The plasmonic imaging system utilized a commercial inverted microscope (Ti microscope, Nikon Co., Japan) with a $100\times$ oil-immersion objective lens ($\text{NA} = 1.49$). Illumination was provided by a 660 nm superluminescent diode to excite surface plasmon resonance. Standard glass slides were replaced with sensing chips modified by self-assembled monolayer with COOH groups. Plasmonic images were recorded by a CCD camera (Pike-032B, Allied Vision Technologies Co., USA) through a $0.46\times$ zoom-out lens (Nikon Co., Japan).

Additionally, we have calculated the significant changes in the RDF of sulfur atom pairs of sulfate over time. To identify the specific structure of the clusters, the RDF diagrams of amorphous phase and perfect crystal of CaSO_4 were compared. As illustrated in **Fig. R4**, the RDF peaks of amorphous calcium sulfate were broad and flattened, while the RDF peaks of crystalline calcium sulfate are sharp and well-defined, exhibiting a degree of long-range order. The simulations performed for the clusters over the period from 0 to 50 ns show that the behavior of the clusters was similar to that of a crystal phase. Therefore, from an atomic-scale perspective, calcium ions and sulfate ions aggregated together, arranging themselves in a crystalline form. These results collectively suggest that the intermediate phase, commonly observed in non-classical nucleation processes, could be excluded in this study.

Fig. R4 The radial distribution function, $g(r)$, of sulfur atom pairs of sulfate with time. CaSO_4 crystal and amorphous phase of CaSO_4 were calculated for comparison.

To address the reviewer’s concern, we have added the two figures (from the new experimental results) into the revised Supplementary Information and related discussion into the revised manuscript as follows:

“Given recent observations of non-classical gypsum nucleation, it is striking to see how gypsum nucleation on foreign surfaces aligns the classical framework. We conducted surface plasmon resonance microscopy experiments, which allow for the nanoscale spatiotemporal identification of nuclei by accurately measuring the refractive index of individual nuclei without interference from background signals. Since the refractive index of crystalline materials is often higher than that of amorphous or liquid materials, the phase transformation of CaSO_4 in the early nucleation process could be identified by this method, which would show a sudden signal change. By tracking the trajectories of CaSO_4 formed on the surface, a time-resolved plasmonic image sequence was captured (Supplementary Fig. 15a). Upon nucleation, a typical scattering pattern was clearly observed. As the time progressed, the plasmonic intensity increased (Supplementary Fig. 15b), indicating CaSO_4 nuclei growth. Notably, the monotonically increased plasmonic intensity excluded a phase change in CaSO_4 . Additionally, we calculated the great changes in the RDF of sulfur atom pairs of sulfate with time. To identify the specific structure of the clusters, the RDF diagrams of amorphous phases and perfect crystal of CaSO_4 were compared. As illustrated in Supplementary Fig. 16, the RDF peaks of amorphous calcium sulfate were broad and flattened, while the RDF peaks of crystalline calcium sulfate were sharp and well-defined, exhibiting a degree of long-range order. Simulations performed for the clusters over the period from 0 to 50 ns show that the behavior of the clusters was similar to that of a crystal phase. Therefore, the intermediate phase, commonly observed in non-classical nucleation process, could be excluded in this study.” (Lines 322-341)

The simulations are carried out at very high supersaturations to reduce the computational time. However, these conditions are not representative of heterogeneous

nucleation and far from the conditions used in the optical microscopy experiments. This discrepancy renders the MD data ineffective for discussing the prevailing heterogeneous nucleation mechanism at low supersaturation, which is the main topic of this manuscript. Moreover, the authors claim a classical nucleation mechanism based on the observation of a critical point where clusters start to grow. Importantly, this is a mere interpretation of the data, with no hard proof provided that this time point is indeed a critical point. Even if true, this observation does not exclude a non-classical nucleation mechanism, which also involves overcoming at least one barrier and thus will have a critical point from which clusters start to grow.

We appreciate your insights and acknowledge the issues you've raised.

1. Supersaturation Discrepancies: Higher concentration solutions were adopted in the original MD simulations mainly because we intended to increase the probability of ion collisions and expedite the simulation process. Additionally, we have conducted MD simulations at lower concentrations. As illustrated in **Fig. R5**, small clusters evolved into large clusters over time, with the cluster structure stabilizing after 40 ns (**Fig. R5**). This process was accompanied by the coordination of calcium ions with sulfate ions and increase in maximum cluster size (**Fig. R6a**). Simultaneously, a notable decrease in the number of free ions was observed (**Fig. Fig. R6b**). Therefore, the nucleation of CaSO_4 in low-concentration solutions resembled that in high-concentration solutions, initiating growth after the critical point and eventually reaching a stable state. However, the nucleation and growth rates were slower and required more time.

We acknowledge that even simulations at low concentrations cannot fully replicate the low concentrations used in experiments. However, the results obtained from simulations at reduced concentrations were consistent with those at higher concentrations, albeit requiring longer simulation times. Thus, it is reasonable to infer that when the simulation concentration was reduced to match the experimental concentration, consistent results would still be achieved. Furthermore, the ion concentration on surface during nucleation process also involved a transition from low to high. Therefore, the MD simulations could be used to discuss the prevailing heterogeneous nucleation mechanism at low supersaturation.

Fig. R5 The dynamic evolution of CaSO_4 cluster with time on $-\text{CH}_3$ surface in the 5 times diluted solution. Only the CaSO_4 clusters were displayed while the rest

were hidden.

Fig. R6 Time evolution of Ca-So bond number and maximum cluster size. (b) Time evolution of cluster number and free ions in the duration of 50 ns.

2. Clarifying the Interpretation of the Critical Point: We have extended the simulation time to 50 ns and found no second critical point. In addition, results from **Fig. R1** and Supplementary Fig. 5 show that these clusters slowly evolved into larger clusters and the size of the top two clusters experienced less change after 25 ns, indicating that these clusters aligned with the CNT framework.

Fig. R1 The dynamic evolution of CaSO₄ cluster with time on -CH₃ surface in the duration of 25-50 ns. Only the CaSO₄ clusters were displayed while the rest were hidden.

3. Classical Nucleation Mechanisms: As discussed in our response to your first comment, we have observed nucleation events using in situ nanoscale spatiotemporal SPRM to definitively exclude the presence of an amorphous phase at the early stages of gypsum nucleation. Furthermore, a detailed examination of the radial distribution function of sulfur atom pairs in sulfate over time ruled out NCNT at the atomic level.

The nucleation rates estimated from the optical microscopy data are difficult to classify due to the low resolution of the microscope, which makes it unclear what is being measured. Consequently, the authors make several assumptions, rendering these data meaningful only to differentiate the effect of the different functionalized surfaces on the crystallization kinetics of gypsum. From the additional experimental details provided in the revisions, it appears that nucleation is also likely occurring in the bulk, but the high flow rates remove these nuclei from the observation cell. This raises the question of what the authors actually measured: the crystallization rate (after substantial growth) of only those crystals that formed on the functionalized surfaces. This measurement is only representative of part of the nucleation occurring. Another important limitation is that observing crystals after significant growth precludes any insight into the true nucleation mechanism and possible post nucleation aggregation. The fact that the nucleation rate dependence on supersaturation can be fitted using an equation derived from the CNT does not provide information about the underlying mechanism. Nucleation rate data for systems like calcium carbonate and gypsum can be readily fitted using CNT, despite both systems being notorious for exhibiting more complex nucleation pathways.

Thank you for your detailed and insightful feedback. We appreciate your thorough review and recognize the need to address the concerns you have raised.

1. Optical Microscopy Data: We acknowledge that there were limitations of optical microscopy due to its low resolution. However, it does not affect the nucleation observation on foreign surfaces as each optically observed crystal was developed from a single nucleus. We assumed that each nucleus, which successfully crossed the energy barrier to achieve critical size, grew into an optically observable crystallite. To say, one crystal, one seed. This assumption is reasonable because the data were collected only within the time interval where crystallite spacing was very large (Supplementary Fig. 3A). Also, during the period of constant nucleation, each crystal consisted of a single, well-faceted rhombohedron. Moreover, the data in Fig. 2 and Supplementary Fig. 4 show that their appearance was linearly dependent on time, which would not be true if nucleation events were not independent of one another.

2. Nucleation Type: We would like to emphasize that, in this work we focused on heterogeneous nucleation on foreign surfaces only. As written in the title, heterogeneous nucleation pathway was regulated by functionalized surfaces. Thus, as the reviewer mentioned, we did measure the crystallization rate (after substantial growth) of only those crystals that were formed on the functionalized surfaces. Nucleation in bulk solution was homogeneous, which is not the focus of our work. The additional experiment provided in the revisions was carried out in a beaker under resting state, not in our flow cell for in situ nucleation observation. Homogeneous and heterogeneous nucleation could occur in the beaker experiment. While in our flow cell, we set the high flow rate to 30 mL/h to prevent possible deposition from homogeneous nuclei. A comparison of these results could ensure that we truly observed heterogeneous nucleation and no nuclei was formed within the bulk of the solution, which subsequently settled on the surface.

3. Nucleation Mechanism: We acknowledge that fitting nucleation rate data with equations derived from CNT does not necessarily indicate the underlying mechanism. Thus, we conducted MD simulations as nucleation on a foreign surface could not be characterized by advanced techniques, such as high-resolution transmission electron microscopy (TEM) and cryo-TEM. These techniques were efficiently employed in homogeneous nucleation process to investigate NCNT (Du et al., 2024; Loh et al., 2017; Van Driessche et al., 2018; Yuwono et al., 2010; Zhu et al., 2021). Additionally, as we mentioned in our response to your first comment, we have observed nucleation events using in situ nanoscale spatiotemporal SPRM to exclude the presence of an amorphous phase at the early stages of gypsum nucleation. Furthermore, a detailed examination of the radial distribution function of sulfur atom pairs in sulfate over time also ruled out NCNT at the atomic level.

Response to Reviewer 3 Comments

I have read the rebuttal and the comments of the comments of other reviewers with interest. I think that the authors did a good job in arguing many of my own and others' doubts. However, I am under the impression that the main sticking point remains.

Thank you for your thorough review and for your encouraging comments on our submission. We appreciate the constructive feedback and the opportunity to further address the main sticking point you have identified.

The authors are trying to bridge, in principle, near-macroscopic optical imaging data with nanoscale MD, to deduce the entire mechanism of growth (and nucleation). This works only within the framework of their interpretation and assumptions (such as one crystal, one seed). This also requires (unfortunately) further experimental evidence. I understand the logics behind this argument, but I disagree with it, because it excludes e.g. the notion of an amorphous phase of CaSO₄.

We highly appreciate your valuable suggestions for enhancing the robustness of our study. To address your concerns, we have conducted in situ nanoscale spatiotemporal surface plasmon resonance microscopy (SPRM) experiments and examined the structural descriptor (radial distribution function of sulfur atom pairs of sulfate). Results reveal that gypsum nucleation on a foreign surface followed a classical pathway.

1. SPRM Experiments: Non-classical nucleation process describes the formation of new phases via the aggregation of individual molecules or atoms. This process involves intermediate species and complex pathways that can lead to the formation of crystalline structures. To investigate whether the crystal was developed from intermediate phase of CaSO₄, SPRM was employed. This technique allows for the high-throughput identification of nuclei by accurately measuring the refractive index of individual nuclei without interference from background signals (Qian et al., 2019; Wu et al., 2024). Since the refractive index of crystalline materials is often higher than that of amorphous or liquid materials, the phase transformation of CaSO₄ in the early nucleation process could be identified by this method, which would show a sudden signal change. By tracking the trajectories of CaSO₄ formed on the surface, a time-resolved plasmonic image sequence was captured (**Fig. R3a**). Upon nucleation of CaSO₄, a typical scattering pattern was clearly observed. As the time progressed, the plasmonic intensity increased (**Fig. R3b**), indicating CaSO₄ nuclei growth. Additionally, the monotonically increased plasmonic intensity excluded a phase change in CaSO₄. This result is consistent with the CNT, which suggests the existence of a critical size that categorizes clusters as either subcritical or supercritical. Thus, under our experimental conditions, intermediate phase of CaSO₄ might not exist as suggested in other studies.

Fig. R3 (a) Snapshot plasmonic images and (b) intensity tracking of CaSO₄ nuclei during the nucleation process. The snapshot when the signal begins to appear was marked as the first frame ($T = 0$ s). Conditions: The plasmonic imaging system utilized a commercial inverted microscope (Ti microscope, Nikon, Japan) with a $100\times$ oil-immersion objective lens ($NA = 1.49$). Illumination was provided by a 660 nm superluminescent diode to excite surface plasmon resonance. Standard glass slides were replaced with sensing chips modified by self-assembled monolayer with COOH groups. Plasmonic images were recorded by a CCD camera (Pike-032B, Allied Vision Technologies) through a $0.46\times$ zoom-out lens (Nikon, Japan).

Additionally, we calculated the significant changes in the RDF of sulfur atom pairs of sulfate over time. To identify the specific structure of the clusters, the RDF diagrams of amorphous phase and perfect crystal of CaSO₄ were compared. As illustrated in **Fig. R4**, the RDF peaks of amorphous calcium sulfate were broad and flattened, while the RDF peaks of crystalline calcium sulfate were sharp and well-defined, exhibiting a degree of long-range order. The simulations performed for the clusters over the period from 0 to 50 ns show that the behavior of the clusters was similar to that of a crystal phase. Therefore, from an atomic-scale perspective, calcium ions and sulfate ions aggregated together, arranging themselves in a crystalline form. These results collectively suggest that the intermediate phase, commonly observed in non-classical nucleation processes, could be excluded in this work.

Fig. R4 The radial distribution function, $g(r)$, of sulfur atom pairs of sulfate with time. CaSO_4 crystal and amorphous phase of CaSO_4 were calculated for comparison.

To address the reviewer’s concern, we have added the two figures (from the new experimental results) into the revised Supplementary Information and related discussion into the revised manuscript as follows:

“Given recent observations of non-classical gypsum nucleation, it is striking to see how gypsum nucleation on foreign surfaces aligns the classical framework. We conducted surface plasmon resonance microscopy experiments, which allow for the nanoscale spatiotemporal identification of nuclei by accurately measuring the refractive index of individual nuclei without interference from background signals. Since the refractive index of crystalline materials is often higher than that of amorphous or liquid materials, the phase transformation of CaSO_4 in the early nucleation process could be identified by this method, which would show a sudden signal change. By tracking the trajectories of CaSO_4 formed on the surface, a time-resolved plasmonic image sequence was captured (Supplementary Fig. 15a). Upon nucleation, a typical scattering pattern was clearly observed. As the time progressed, the plasmonic intensity increased (Supplementary Fig. 15b), indicating CaSO_4 nuclei growth. Notably, the monotonically increased plasmonic intensity excluded a phase change in CaSO_4 . Additionally, we calculated the great changes in the RDF of sulfur atom pairs of sulfate with time. To identify the specific structure of the clusters, the RDF diagrams of amorphous phases and perfect crystal of CaSO_4 were compared. As illustrated in Supplementary Fig. 16, the RDF peaks of amorphous calcium sulfate were broad and flattened, while the RDF peaks of crystalline calcium sulfate were sharp and well-defined, exhibiting a degree of long-range order. Simulations performed for the clusters over the period from 0 to 50 ns show that the behavior of the clusters was similar to that of a crystal phase. Therefore, the intermediate phase, commonly observed in non-classical nucleation process, could be excluded in this study.” (Lines 322-341)

The argument that other methods are ex situ is not true: methods such as Raman spectroscopy or X-ray scattering can be applied very much in situ. So for instance, if

we stick to the heterogenous nucleation aspects, why not use grazing-incidence SAXS and WAXS to support these findings? In particular, the answer which needs to be answered is whether we deal here with a single stage process (i.e. classical) or a multi-stage one. Is there any amorphous phase present? Dissolution-precipitation of such a phase would also lead to a growth of crystal nuclei which would look, in later stages, like a single stage process in optical microscopy. For instance the authors could measure in situ with Raman spectroscopy changes in the intensity of a liquid sulfate band at $\sim 980\text{ cm}^{-1}$, correlate it with the concentration of this species, and extract changes in the sulfate profile in correlation with the growth rate of crystals from optical microscopy. Ideally, these trends should align.

We acknowledge that several in situ methods, such as Raman spectroscopy and X-ray scattering, can provide real-time insights into the nucleation and growth processes. We agree that incorporating these techniques could strengthen our study and address the key questions regarding the nucleation mechanism. We have tried Raman spectroscopy, but failed to obtain solid data.

1. Raman Spectroscopy: We have tried to apply in situ Raman spectroscopy to monitor changes in the intensity of the liquid sulfate band at $\sim 980\text{ cm}^{-1}$. The signal was negligible (**Fig. R7**), which was ascribed to the low concentration used in our experiments (0.05 mmol/L). The signal would be obvious only under high concentration. 0.2 mol/L-1.1 mol/L was adopted in the literature (Dongliang et al., 2015).

Fig. R7 Raman characterization of sodium sulfate solution. Conditions: Raman spectroscopic characterization was conducted with a He-Ne 532 nm excitation laser (0.544 mW), a 600 lines/mm grating, and an Olympus LMPlan FLN 50 × objective (N.A. 0.50).

2. Grazing-Incidence XRD: We ended the nucleation experiment at 5, 10, and 30 s, and took the substrates for grazing-incidence XRD characterization. No signal was obtained. The absence of crystalline peaks for nuclei at the substrate could be explained by two reasons: First, in the early time of nucleation, the nuclei is very small and separately distributed on the surface of substrate. As a consequence, the number might be insufficient to generate a noticeable peak intensity. Second, the newly formed particles can be amorphous and therefore do not exhibit XRD peaks, which could be ruled out based on our simulation results (**Fig. R4**). Additionally, we have re-analyzed the studies about nucleation

mechanism using GISAXS or GIWAXS. As shown in Table R1, the multi-step nucleation processes were found in homogeneous aqueous solution. However, heterogeneous nucleation on a solid foreign surface often fitted well with CNT, which was consistent with our results.

Table R1. Nucleation mechanism using GISAXS or GIWAXS from literatures

Item	System	Technique	Main conclusion	Ref.
1	gypsum homogeneous aqueous solution	In situ SAXS/WAXS	The formation of gypsum proceeds through a complex four-stage process. The reaction starts through the fast formation of well-defined, primary species of <3 nm in length (stage I), followed in stage II by their arrangement into domains.	(Stawski et al., 2016)
2	calcium phosphate homogeneous aqueous solution	In situ SAXS/WAXS	Amorphous particle formed in the first stages of reaction. As the reaction time progresses, amorphous particles evolve into crystalline ones, whose kinetics of crystal growth are controlled by temperature and carboxylate ions.	(Siliqi et al., 2023)
3	Calcite nucleation on quartz	In situ SAXS	heterogeneous nucleation of calcium carbonate is favored on quartz by calculating the interfacial free energy using classical nucleation theory	(Fernandez-Martinez et al., 2013)
4	The effects of salinity on CaCO ₃ nucleation	In situ GISAXS	High salinity triggered faster nucleation based on	(Li and Jun, 2019)

	on quartz		interfacial energy and kinetic factor. These important parameters are obtained based on the nucleation rate equation that does not convey the possible pathway of forming a metastable phase before the nuclei transform to a stable phase.	
5	Iron(III) (hydr)oxide nucleation on quartz	In situ GISAXS	By quantitative analyses based on classical nucleation theory, α' was obtained to be 34.6 mJ/m ² and E_a was quantified as 32.8 kJ/mol.	(Wu et al., 2020)
6	The effect of surface functional groups on iron (hydr)oxide heterogeneous nucleation	In situ GISAXS	This study, for the first time, applied CNT to obtain the quantitative thermodynamics of iron (hydr)oxide heterogeneous nucleation.	(Chou et al., 2023)

Anyhow, I think it is a very good work, and I would like to see it published. But only with more experimental evidence.

We fully understand the importance of providing robust experimental evidence to support our findings. In response to your suggestion, we have conducted additional experiments and simulations to further substantiate our results (**Figs. R1-7**). Correspondingly, we have included the new experimental data and discussion in the revised manuscript to ensure that our conclusions are well-supported and comprehensive (**Lines 322-341**).

References

- Chou, P.-I., Ghim, D., Gupta, P., Singamaneni, S., Lee, B. and Jun, Y.-S. 2023. Surface Functional Groups Affect Iron (Hydr)oxide Heterogeneous Nucleation: Implications for Membrane Scaling. *Environmental Science & Technology* 57(30), 11056-11066.
- Dongliang, Z.H.U., Ziang, Z.H.U., Junyi, P.A.N., Junying, D. and Pei, N.I. 2015. Raman Micro-Spectroscopic Study of Sulfate Ion in the System Na₂SO₄ – H₂O. *Acta Geologica Sinica - English Edition* 89(3), 887-893.
- Du, J.S., Bae, Y. and De Yoreo, J.J. 2024. Non-classical crystallization in soft and organic materials. *Nature Reviews Materials* 9(4), 229-248.
- Fernandez-Martinez, A., Hu, Y., Lee, B., Jun, Y.-S. and Waychunas, G.A. 2013. In Situ Determination of Interfacial Energies between Heterogeneously Nucleated CaCO₃ and Quartz Substrates: Thermodynamics of CO₂ Mineral Trapping. *Environmental Science & Technology* 47(1), 102-109.
- Li, Q. and Jun, Y.-S. 2019. Salinity-Induced Reduction of Interfacial Energies and Kinetic Factors during Calcium Carbonate Nucleation on Quartz. *The Journal of Physical Chemistry C* 123(23), 14319-14326.
- Loh, N.D., Sen, S., Bosman, M., Tan, S.F., Zhong, J., Nijhuis, C.A., Král, P., Matsudaira, P. and Mirsaidov, U. 2017. Multistep nucleation of nanocrystals in aqueous solution. *Nature Chemistry* 9(1), 77-82.
- Qian, C., Wu, G., Jiang, D., Zhao, X., Chen, H.-B., Yang, Y. and Liu, X.-W. 2019. Identification of Nanoparticles via Plasmonic Scattering Interferometry. *Angew. Chem. Int. Ed.* 58(13), 4217-4220.
- Siliqi, D., Adamiano, A., Ladisa, M., Giannini, C., Iafisco, M. and Degli Esposti, L. 2023. Formation of calcium phosphate nanoparticles in the presence of carboxylate molecules: a time-resolved in situ synchrotron SAXS and WAXS study. *CrystEngComm* 25(4), 550-559.
- Stawski, T.M., van Driessche, A.E.S., Ossorio, M., Diego Rodriguez-Blanco, J., Besselink, R. and Benning, L.G. 2016. Formation of calcium sulfate through the aggregation of sub-3 nanometre primary species. *Nature Communications* 7(1), 11177.
- Van Driessche, A.E.S., Van Gerven, N., Bomans, P.H.H., Joosten, R.R.M., Friedrich, H., Gil-Carton, D., Sommerdijk, N.A.J.M. and Sleutel, M. 2018. Molecular nucleation mechanisms and control strategies for crystal polymorph selection. *Nature* 556(7699), 89-94.
- Wu, G., Lv, W.-L., Qian, C. and Liu, X.-W. 2024. High-Throughput Identification of Single Nanoparticles via Electrochemically Assisted High-Resolution Plasmonic Scattering Interferometric Microscopy. *Nano Lett.* 24(20), 6124-6130.
- Wu, X., Lee, B. and Jun, Y.-S. 2020. Interfacial and Activation Energies of Environmentally Abundant Heterogeneously Nucleated Iron(III) (Hydr)oxide on Quartz. *Environmental Science & Technology* 54(19), 12119-12129.

- Yuwono, V.M., Burrows, N.D., Soltis, J.A. and Penn, R.L. 2010. Oriented Aggregation: Formation and Transformation of Mesocrystal Intermediates Revealed. *Journal of the American Chemical Society* 132(7), 2163-2165.
- Zhu, G., Sushko, M.L., Loring, J.S., Legg, B.A., Song, M., Soltis, J.A., Huang, X., Rosso, K.M. and De Yoreo, J.J. 2021. Self-similar mesocrystals form via interface-driven nucleation and assembly. *Nature* 590(7846), 416-422.

Response to Reviewer 1's Comments

Overall, the authors have responded my concern properly. And the results of Guan et al. provide new insights into heterogeneous nucleation mechanism of gypsum, and is helpful for exploration of the nucleation mechanism in solution. So I think the manuscript of Guan et al may be accepted for publication.

We highly appreciate the reviewer's valuable time and efforts on our manuscript.

Response to Reviewer 2's Comments

The authors have provided additional data; however, the primary issue raised during the initial round of revisions by all reviewers remains unresolved. Based on the available data, it is still not possible to definitively determine whether the nucleation pathway follows a classical or non-classical trajectory.

The first new data set was obtained through surface plasmon resonance microscopy (SPRM) experiments. The primary finding from these experiments is that, upon nucleation of CaSO₄, a characteristic scattering pattern was observed. Over time, the plasmonic intensity increased (Fig. R3b), which the authors interpret as evidence of CaSO₄ nuclei growth. They suggest that this result aligns with Classical Nucleation Theory (CNT), which proposes the existence of a critical size distinguishing subcritical from supercritical clusters. However, this reasoning has two major flaws. First, if only growing clusters are observed, how can one confirm the existence of a critical cluster size? To establish this, one should also observe dissolving clusters. Second, the observation of growing clusters does not exclude non-classical nucleation, as cluster growth is also expected in non-classical pathways. Additionally, several other issues arise. The spatial resolution of the technique is not clearly stated, leaving the size of the detected particles ambiguous. Furthermore, how was the refractive index difference between crystalline and disordered CaSO₄ phases determined?

We appreciate your insightful comments and have made revisions to address your concerns. Our response is given below:

1. Critical Cluster Size and Dissolving Clusters: We acknowledge that the observation of growing clusters only does not conclusively confirm the existence of a critical cluster size as proposed by CNT. We agree that, to definitively establish the critical size, the observation of both growing and dissolving clusters would be ideal. Unfortunately, due to the limitations in surface detection of SPRM, observing dissolution of clusters in the bulk solution is not feasible at the present stage. Instead, we propose that our observations of growing clusters are consistent with CNT, but do not exclude other nucleation pathways. In this way, a more balanced interpretation has been provided in revised manuscript as follows (**Lines 372-382**).

“Our findings align with CNT, providing a foundation for a more comprehensive and precise grasp of heterogeneous gypsum nucleation on surfaces, a critical aspect in the effective management of mineral scaling in various industrial processes (e.g., membrane desalination). Nonetheless, it is important to recognize the limitations of our experimental techniques—specifically, the inability to directly observe critical cluster sizes, dissolved clusters, or crystalline structures that remain internally amorphous. It is possible that certain non-classical processes may have evaded our detection. Consequently, deeper exploration into the early stages of gypsum’s complex growth patterns on diverse surfaces, using advanced characterization techniques operating at the sub-nano scale or even smaller, will be essential to comprehensively capture these potential alternative pathways.”

2. Non-Classical Nucleation Pathways: Amorphous phase is commonly found in non-classical nucleation pathways. To rule out amorphous phase, we have conducted control experiments with a system known to form a long-lived amorphous calcium phase (i.e., calcium phosphate). We tracked the trajectories of amorphous calcium phosphate formed on carboxyl-modified substrates and captured a time-resolved plasmonic image sequence (**Fig. R1a**). The amorphous calcium phosphate exhibited a characteristic scattering pattern, indicating that SPRM could detect amorphous phase. However, its intensity was weak and remained unchanged as the time progressed (**Fig. R1b**), suggesting the stability of the amorphous, which is consistent with its long-live nature. By contrast, this result differs from gypsum nucleation, in which intensity gradually increases over time after nuclei formation on the substrate (**Supplementary Fig. 15**).

We noticed the obvious differences in the brightness and width of the central region of the scattering pattern between crystalline CaSO_4 and amorphous calcium phosphate. A further analysis of the phase parameter (ψ) of the scattering patterns, based on previous studies (He et al., *ACS Nano*, 2024, 18(13), 9704-9712; Qian et al., *Angew. Chem. Int. Ed.*, 2019, 58(13), 4217-4220; Wu et al., *Nano Lett.*, 2022, 22(11), 4383-4391), reveals that crystalline CaSO_4 has a ψ value of 0.13, while amorphous calcium phosphate exhibits a ψ value of -0.35. **This result demonstrates that SPRM was capable of distinguishing between amorphous and crystalline phases.** These findings together confirm that no amorphous phase was detected in the gypsum nucleation system during our SPRM tracking, as supported by the gradually increased intensity and stable ψ values over time, thus excluding the presence of an amorphous form.

Methods for amorphous calcium phosphate: it was prepared in Tris-buffered saline, following the established protocols (Habraken et al., *Nat. Commun.*, 2013, 4(1), 1507; Song et al., *Cryst. Growth Des.*, 2023, 23(10), 7150-7158). The reactions occurred in a solution containing 50 mM Tris and 50 mM NaCl dissolved in ultrapure water (Milli-Q standard, 25 °C), with the pH adjusted to 7.4 ± 0.05 using 1 M HCl. A 10 mM calcium stock solution was prepared by dissolving CaCl_2 in this Tris-buffered saline, followed by adjusting the pH to 7.40 ± 0.05 using 0.1 M NaOH. Additionally, a 10 mM phosphate stock solution was prepared by dissolving K_2HPO_4 in Tris-buffered saline, and the pH was similarly adjusted to 7.40 ± 0.05 with 1 M HCl. A total of 294 μL of the 10 mM calcium solution was introduced into the chamber for baseline correction of the SPRM, followed by dosing 206 μL of the 10 mM phosphate solution to generate long-lived amorphous calcium phosphate for SPRM detection.

Fig. R1 (a) Snapshot plasmonic images and (b) intensity tracking of amorphous calcium phosphate on substrate functionalized with carboxyl groups. The snapshot when the signal began to appear was marked as the first frame ($T = 0$ s). Conditions: The plasmonic imaging system utilized a commercial inverted microscope (Ti microscope, Nikon Co., Japan) with a $100\times$ oil-immersion objective lens ($NA = 1.49$). Illumination was provided by a 660 nm superluminescent diode to excite surface plasmon resonance. Standard glass slides were replaced with sensing chips modified by self-assembled monolayer with COOH groups. Plasmonic images were recorded by a CCD camera (Pike-032B, Allied Vision Technologies Co., USA) through a $0.46\times$ zoom-out lens (Nikon Co., Japan).

3. Spatial Resolution of SPRM: As an optical microscopy technique, SPRM is limited by diffraction limits (250 nm in spatial resolution). However, **as an advanced interface analysis technique, it boasts exceptional sensitivity for detecting variations in interface refractive indices, allowing for real-time, label-free detection of dynamic processes in solution** (Wu et al., *Nat. Commun.*, 2023, 14(1), 4194). It is capable of distinguishing materials with closely similar refractive indices (Wu et al., *Nano Lett.*, 2024, 24(20), 6124-6130), detecting nanoparticles as small as a

few nanometers (Qian et al., *Angew. Chem. Int. Ed.*, 2019, 58(13), 4217-4220), and identifying ions with an extraordinarily low detection limit of 1 attomolar (Zhang et al., *Nano Lett.*, 2024). Furthermore, we used varying concentrations of CaCl₂ as a case study to assess SPRM's response. As illustrated in **Fig. R2** and **Table R1**, SPRM exhibited exceptional performance with a wide detection range (0.01 to 1 M) and a low detection limit of 0.01 M CaCl₂. Notably, the refractive index of 0.01 M CaCl₂ is 1.332, which is slightly higher than that of water (refractive index is 1.33), with water serving as the background.

Table R1. The refractive indices corresponding to different concentrations of CaCl₂ (Wang et al., *Sens. Actuators B Chem.*, 2015, 210, 649-655)

Concentration of CaCl ₂ (M)	0	0.01	0.05	0.5	1
Refractive Indices	1.33	1.3329	1.3340	1.3456	1.3585

Fig. R2 The intensity response of CaCl₂ solutions with different refractive indices in SPRM.

For clarification, we have supplemented the spatial resolution in the revised manuscript as follows:

This setup provides exceptional sensitivity for detecting variations in interface refractive indices, allowing for real-time, label-free detection of dynamic processes in solution. (Lines 413-415)

4. Refractive Index Difference Between Crystalline and Disordered Phases:

The refractive index of a material, denoted as n , is defined as the ratio of the speed of light in a vacuum to its speed in the material:

$$n = \frac{c}{c_m} \quad 1$$

where c is the speed of light in a vacuum, and c_m the speed of light in the material.

Thus, the refractive index is intrinsically related to the material's density, degree of ordering, and morphology (Al-Ani, *Iraqi Journal of Applied Physics*, 2008, 4, 17-23; Kofman et al., *Astrophys. J.*, 2019, 875(2), 131). Generally, materials with a higher

degree of ordering and density tend to exhibit a greater refractive index, as the well-ordered crystal structures allow for more efficient interaction with light, resulting in increased light wave refraction. For example, thermal annealing of amorphous TiO₂ coating to achieve a crystalline form enhances their refractive index (Wang et al., *Materials*, 2013, 6(7), 2819-2830; Yao et al., *Surf. Eng.*, 2009, 25(3), 257-260). We have incorporated the relevant references into the manuscript.

Since the refractive index of crystalline materials is often higher than that of amorphous or liquid materials^{65, 66} (**Lines 327**)

The second new piece of information is the Radial Distribution Function analysis of the original simulation data. This analysis shows that the clusters lack long-range order, extending no further than 8 Å, which corresponds well with the Pair Distribution function(PDF) analysis of scattering data collected during the early stages of gypsum precipitation (*J. Phys. Chem. C* 2019, 123, 37, 23151–23158). Additionally, simply by examining the simulation snapshots provided by the authors, it is evident that the clusters are not well-defined or crystalline. Moreover, the simulations were conducted at high supersaturation levels, which are not representative of the typical range of supersaturations where heterogeneous nucleation is expected. Overall, these findings do not provide sufficient evidence to confirm or refute classical nucleation in the case of heterogeneous nucleation.

We sincerely thank the reviewer for the valuable comments. Our detailed response is given below:

1. Radial Distribution Function (RDF) and Long-range Order: While our RDF profiles, based on the supplementary simulation data, look similar to Figure 2 in the reference (*J. Phys. Chem. C* 2019, 123, 37, 23151–23158), there are notable differences in our findings: a) The reference in Figure 2 in the reference, which presents the Pair Distribution Function (PDF) analysis of CaSO₄, and emphasizes that clusters at the early stages lack long-range order, as indicated by the absence of peaks beyond 8 Å. By contrast, our RDF results, focusing on S-S pairs, revealed distinct peaks beyond 10 Å. b) The PDF curve (**Fig. R3a**, Figure 2 in the reference) from the MD simulation includes constraints, such as ~0.5 nm distances and nanometer-scale structures. Template clusters were built by stacking primary motifs containing Ca-Ca bond distances, then optimized using the randomized simplex method to align the simulated PDF with experimental G(r) data. This introduces regularity, reflecting some crystalline characteristics, making the system not entirely amorphous. c) Upon carefully revisiting this reference, Figure 4 depicts the RDF of S-S pairs (**Fig. R3b**), where two peaks are observed at approximately 5 and 7 Å. Interestingly, this outcome closely resembles the amorphous phase in our S-S RDF profiles, where two initial peaks are located at 4.55 and 6.95 Å. d). In our work, we have compared the RDF profiles of both the amorphous phase and gypsum crystal to those of the solution systems. As illustrated in **supplementary Fig. 16**, the RDF peak for the amorphous phase was broader and flatter compared to that of the gypsum crystal and solution systems, indicating the absence of long-range order. By contrast, the gypsum crystal RDF showed a well-defined multi-

peak structure (more than five peaks) with distinct long-range order. Additionally, in the enlarged view of the solution system's RDF profiles (**Fig. R4**, highlighted with yellow dashed lines), no fewer than five peaks were discernible. Although the positions of these peaks differed slightly from those in the gypsum crystal, their overall character more closely resembled that of a crystalline phase, rather than an amorphous one.

Fig. R3 (a) Figure 2 in the reference shows pair distribution function analysis of CaSO_4 . (b) Figure 4 in the reference depicts the radial distribution function of S-S pairs.

Fig. R4 The locally enlarged figure of supplementary Fig. 16, the yellow dashed lines are the approximate positions of peak at 50 ns.

2. Cluster Definition and Crystallinity from Simulation Snapshots: We understand the reviewer's concern regarding the apparent amorphous shape of clusters in the simulation snapshots. This observation could be partially attributed to the presence of water molecules, which influenced the surface ions of the clusters, leading to the visual appearance of disorder. However, when analyzing through the RDF profiles, it became evident that the inner ions within the clusters had started to organize into a crystalline structure. This evolving crystallinity is further supported by the comparison of RDF profiles over time (from 5 ns to 50 ns, **Fig. R4**), which demonstrates that the clusters were progressively moving toward a higher degree of order consistent with gypsum crystals.

3. High Supersaturation Levels in Simulations: We agree with the reviewer that the supersaturation levels in our simulations exceeded those typically encountered

under experimental conditions. This decision was made to accelerate nucleation events within a feasible simulation timescale, a common approach in molecular dynamics simulations due to spatial and temporal limitations (Li et al., *Nat. Commun.*, 2021, 12(1), 4954; Nicholas et al., *Nat. Chem.*, 2024, 16(1), 36-41). If the solution concentration was reduced to match experimental levels, the number of ions present in the simulation would be insufficient, and increasing the volume to accommodate more ions would largely raise computational costs. We have provided supplementary data in our last response to show that at lower concentrations, nucleation proceeded similarly, indicating that concentration did not have a great impact on the nucleation stage.

Response to Reviewer 3's Comments

Thank you for addressing my previous comments and for putting in the extra effort with the SPRM experiments. I think SPRM is a good approach, but I am still not fully convinced that it can completely rule out the presence of an amorphous phase or a dense liquid precursor (thereby resolving NCNT vs CNT). I would say that the challenge is that proving something does not exist is harder than proving it does. My concern is that an amorphous calcium sulfate phase, or its dense liquid precursor equivalent, might have a refractive index too close to the surrounding solution. If that is the case, SPRM might not be able to detect it. Or am I missing the point? Moreover, the concentration or population of these precursor phases or pre-nucleation clusters could be very low. How would you account for that in your measurements, especially since detecting low concentrations with SPRM could be tricky (right)? To address this, I think it would be helpful if you ran some control experiments using a system that is known to form long-lived amorphous phases, e.g. calcium phosphate at similar concentrations. This would give a better idea of how well SPRM can pick up amorphous phases and would add more confidence to your conclusions. Also, the assumption is that a large feature in Fig. R3 is always a growing crystal? This would have to be verified. Following the notion of particle mediated growth of calcium sulfate, one of the concepts is that we deal with a proto-structure, where "building units" aggregate into a large feature (an aggregate), where the crystallisation takes place from inside out. It has been shown for some protein crystallisation events that this large aggregate may externally look like a crystal, but be still amorphous.

Many thanks for your thoughtful and detailed feedback. We appreciate your recognition of our efforts with the SPRM experiments and understand your concerns regarding the potential limitations of the technique. Below we would like to address each of the points raised:

1. **Detection of Amorphous Phases or Dense Liquid Precursors:** We fully agree that proving the absence of an amorphous phase or a dense liquid precursor is inherently more challenging than confirming their presence. To address your concern about the refractive index being too close to the surrounding solution, potentially limiting SPRM's ability to detect amorphous calcium sulfate phases or their precursors, we have conducted an experiment using 0.01 M CaCl₂, whose refractive index is 1.3329 (**Table R1**), which is slightly higher than that of water (refractive index is 1.33), with water serving as the background. A noticeable signal enhance was obtained once 0.01 M CaCl₂ added into the water (**Fig. R2**). This result demonstrates that SPRM exhibited an exceptional sensitivity to distinguish the substances with refractive index close to the surrounding solution.

Table R1. The refractive indices corresponding to different concentrations of CaCl₂ (Wang et al., 2015)

Concentration of CaCl ₂ (M)	0	0.01	0.05	0.5	1
Refractive Indices	1.33	1.3329	1.3340	1.3456	1.3585

Fig. R2 The intensity response of CaCl_2 solutions with different refractive indices in SPRM.

2. The effect of Concentration on SPRM Detection: First, we would like to clarify that **SPRM, as an advanced interface analysis technique, boasts exceptional sensitivity for detecting variations in interface refractive indices, allowing for real-time, label-free detection of dynamic processes in solution** (Wu et al., *Nat. Commun.*, 2023, 14(1), 4194). It is capable of distinguishing materials with closely similar refractive indices (Wu et al., *Nano Lett.*, 2024, 24(20), 6124-6130) and detecting ions with an extraordinarily low detection limit of 1 attomolar (Zhang et al., *Nano Lett.*, 2024). Furthermore, we used varying concentrations of CaCl_2 as a case study to assess SPRM's response. As illustrated in **Fig. R2 and Table R1**, SPRM exhibited exceptional performance with a wide detection range (0.01 to 1 M) and a low detection limit of 0.01 M CaCl_2 . Notably, the refractive index of 0.01 M CaCl_2 is 1.332, which is slightly higher than that of water (refractive index is 1.33), with water serving as the background.

3. Control Experiments with Calcium Phosphate: We appreciate your suggestion about the use of control experiments with a system known to form long-lived amorphous calcium phases (i.e., calcium phosphate). We concur that such controls would enhance the robustness of our conclusions and have conducted such experiments. We tracked the trajectories of amorphous calcium phosphate formed on carboxyl-modified substrates and captured a time-resolved plasmonic image sequence (**Fig. R1a**). The amorphous calcium phosphate exhibited a characteristic scattering pattern, indicating that SPRM could detect amorphous phase. However, its intensity was weak and remained unchanged as the time progressed (**Fig. R1b**), suggesting the stability of the amorphous, which is consistent with its long-live nature. By contrast, this result differs from gypsum nucleation, in which intensity gradually increased over time after nuclei formation on the substrate (**Supplementary Fig. 15**).

We noticed the obvious differences in the brightness and width of the central region of the scattering pattern between crystalline CaSO_4 and amorphous calcium phosphate. A further analysis of the phase parameter (ψ) of the scattering patterns, based on previous studies (He et al., *ACS Nano*, 2024, 18(13), 9704-9712; Qian et al., *Angew.*

Chem. Int. Ed., 2019, 58(13), 4217-4220; Wu et al., *Nano Lett.*, 2022, 22(11), 4383-4391), reveals that crystalline CaSO_4 had a ψ value of 0.13, while amorphous calcium phosphate exhibited a ψ value of -0.35. This result demonstrates that SPRM was capable of distinguishing between amorphous and crystalline phases. These findings together confirm that no amorphous phase was detected in the gypsum nucleation system during our SPRM tracking, as evidenced by the gradually increased intensity and stable ψ values over time, thus excluding the presence of an amorphous form.

Methods for amorphous calcium phosphate: it was prepared in Tris-buffered saline, following the established protocols (Habraken et al., *Nat. Commun.*, 2013, 4(1), 1507; Song et al., *Cryst. Growth Des.*, 2023, 23(10), 7150-7158). The reactions occurred in a solution containing 50 mM Tris and 50 mM NaCl dissolved in ultrapure water (Milli-Q standard, 25 °C), with the pH adjusted to 7.4 ± 0.05 using 1 M HCl. A 10 mM calcium stock solution was prepared by dissolving CaCl_2 in this Tris-buffered saline, followed by adjusting the pH to 7.40 ± 0.05 using 0.1 M NaOH. Additionally, a 10 mM phosphate stock solution was prepared by dissolving K_2HPO_4 in Tris-buffered saline, and the pH was similarly adjusted to 7.40 ± 0.05 with 1 M HCl. A total of 294 μL of the 10 mM calcium solution was introduced into the chamber for baseline correction of the SPRM, followed by dosing 206 μL of the 10 mM phosphate solution to generate long-lived amorphous calcium phosphate for SPRM detection.

Fig. R1 (a) Snapshot plasmonic images and (b) intensity tracking of amorphous calcium phosphate on substrate functionalized with carboxyl groups. The snapshot when the signal began to appear was marked as the first frame ($T = 0$ s). Conditions: The plasmonic imaging system utilized a commercial inverted microscope (Ti microscope, Nikon Co., Japan) with a 100 \times oil-immersion objective lens ($NA = 1.49$). Illumination was provided by a 660 nm superluminescent diode to excite surface plasmon resonance. Standard glass slides were replaced with sensing chips modified by self-assembled monolayer with COOH groups. Plasmonic images were recorded by a CCD camera (Pike-032B, Allied Vision Technologies Co., USA) through a $0.46 \times$ zoom-out lens (Nikon Co., Japan).

4. Large Features in Fig. R3 and Particle-Mediated Growth: We acknowledge the limitations of SPRM in detecting protein-like crystallization events, in which appeared crystalline externally, but remained amorphous internally. Consequently, our experimental observations and simulations align with classical nucleation pathway (CNT), but expressed reservations regarding alternative pathways, such as non-classical nucleation. Accordingly, we have refined our discussion to reflect such balance in the revised manuscript as follows:

Our findings align with CNT, providing a foundation for a more comprehensive and precise grasp of heterogeneous gypsum nucleation on surfaces, a critical aspect in the effective management of mineral scaling in various industrial processes (e.g., membrane desalination). Nonetheless, it is important to recognize the limitations of our experimental techniques—specifically, the inability to directly observe critical cluster sizes, dissolved clusters, or crystalline structures that remain internally amorphous. It is possible that certain non-classical processes may have evaded our detection. Consequently, deeper exploration into the early stages of gypsum’s complex growth patterns on diverse surfaces, using advanced characterization techniques operating at the sub-nano scale or even smaller, will be essential to comprehensively capture these potential alternative pathways. (Lines 372-382)

Regarding MD simulations and new plots: I am slightly puzzled about the emphasis you placed on the sulphur RDFs rather than the calcium. While the sulphur RDFs are useful in identifying sulfate structures, the coordination environment around calcium could provide more direct evidence of how clusters form, stabilize, and potentially transition into crystalline phases. Was there a specific reason that sulphur RDFs were the focus?

Thanks a lot for the reviewer’s helpful comments. During the actual nucleation process, calcium ions interact strongly with the oxygen atoms in sulfate. The binding affinity of calcium ions to oxygen atoms is exceptionally high, resulting in a pronounced first peak in the RDF curve. Sulfur atoms, being the central atoms of the sulfate group, do not directly interact with calcium ions; rather, their relative positions shift passively as a consequence of the interaction between calcium ions and the oxygen atoms of sulfate. Such passive positional shift is more effective in distinguishing the structural features of crystalline versus amorphous phases. Furthermore, we have also

computed the RDF curves for Ca-S interactions for comparison, as depicted in Fig. R5. It is evident that the ion clusters in the solution exhibited a closer resemblance to the gypsum crystal structure.

Fig. R5 The radial distribution function, $g(r)$, of calcium-sulfur (Ca-S) pairs with time. gypsum crystal and amorphous phase were calculated for comparison. The inset is the locally enlarged figure, the yellow dashed lines are the approximate positions of peak at 50 ns.

References

- Al-Ani, S.K. 2008. Methods of determining the refractive index of thin solid films (article review). *Iraqi Journal of Applied Physics* 4, 17-23.
- Habraken, W.J.E.M., Tao, J., Brylka, L.J., Friedrich, H., Bertinetti, L., Schenk, A.S., Verch, A., Dmitrovic, V., Bomans, P.H.H., Frederik, P.M., Laven, J., van der Schoot, P., Aichmayer, B., de With, G., DeYoreo, J.J. and Sommerdijk, N.A.J.M. 2013. Ion-association complexes unite classical and non-classical theories for the biomimetic nucleation of calcium phosphate. *Nature Communications* 4(1), 1507.
- He, Y.-F., Yang, S.-Y., Lv, W.-L., Qian, C., Wu, G., Zhao, X. and Liu, X.-W. 2024. Deep-Learning Driven, High-Precision Plasmonic Scattering Interferometry for Single-Particle Identification. *ACS Nano* 18(13), 9704-9712.
- Kofman, V., He, J., Loes ten Kate, I. and Linnartz, H. 2019. The Refractive Index of Amorphous and Crystalline Water Ice in the UV–vis. *The Astrophysical Journal* 875(2), 131.
- Li, C., Liu, Z., Goonetilleke, E.C. and Huang, X. 2021. Temperature-dependent kinetic pathways of heterogeneous ice nucleation competing between classical and non-classical nucleation. *Nature Communications* 12(1), 4954.
- Nicholas, T.C., Stones, A.E., Patel, A., Michel, F.M., Reeder, R.J., Aarts, D.G.A.L., Deringer, V.L. and Goodwin, A.L. 2024. Geometrically frustrated interactions drive structural complexity in amorphous calcium carbonate. *Nature Chemistry* 16(1), 36-41.
- Qian, C., Wu, G., Jiang, D., Zhao, X., Chen, H.B., Yang, Y. and Liu, X.W. 2019. Identification of Nanoparticles via Plasmonic Scattering Interferometry. *Angew. Chem. Int. Ed. Engl.* 58(13), 4217-4220.
- Song, H., Cai, M., Fu, Z. and Zou, Z. 2023. Mineralization Pathways of Amorphous Calcium Phosphate in the Presence of Fluoride. *Crystal Growth & Design* 23(10), 7150-7158.
- Wang, L., Zhao, C., Duits, M.H.G., Mugele, F. and Siretanu, I. 2015. Detection of ion adsorption at solid–liquid interfaces using internal reflection ellipsometry. *Sensors and Actuators B: Chemical* 210, 649-655.
- Wang, X., Wu, G., Zhou, B. and Shen, J. 2013. Optical Constants of Crystallized TiO₂ Coatings Prepared by Sol-Gel Process. *Materials* 6(7), 2819-2830.
- Wu, G., Lv, W.-L., Qian, C. and Liu, X.-W. 2024. High-Throughput Identification of Single Nanoparticles via Electrochemically Assisted High-Resolution Plasmonic Scattering Interferometric Microscopy. *Nano Lett.* 24(20), 6124-6130.
- Wu, G., Qian, C., Lv, W.-L., Zhao, X. and Liu, X.-W. 2023. Dynamic imaging of interfacial electrochemistry on single Ag nanowires by azimuth-modulated plasmonic scattering interferometry. *Nature Communications* 14(1), 4194.
- Wu, G., Zhou, X., Lv, W.L., Qian, C. and Liu, X.W. 2022. Real-Time Plasmonic Imaging of the Compositional Evolution of Single Nanoparticles in Electrochemical Reactions. *Nano Lett.* 22(11), 4383-4391.
- Yao, J.K., Huang, H.L., Ma, J.Y., Jin, Y.X., Zhao, Y.A., Shao, J.D., He, H.B., Yi, K., Fan, Z.X., Zhang, F. and Wu, Z.Y. 2009. High refractive index TiO₂ film

deposited by electron beam evaporation. Surf. Eng. 25(3), 257-260.
Zhang, W., Pan, X., Yan, J., Liu, L., Nie, A., Cheng, Y., Wen, F., Mu, C., Zhai, K., Xiang, J., Wang, B., Xue, T. and Liu, Z. 2024. High-Active Surface of Centimeter-Scale β -In₂S₃ for Attomolar-Level Hg²⁺ Sensing. Nano Lett.

Response to Reviewer 2's Comments

I would like to begin by acknowledging the commendable effort of the authors in addressing the reviewers' comments, including the addition of new experimental data. This clearly reflects their motivation to generate new insights. At this stage, I think we can all agree that there is insufficient evidence to determine which pathway prevails during heterogeneous gypsum nucleation under the tested conditions, and additional experiments beyond the scope of this study are required to resolve this issue. As a result, the authors have removed most of their previous claims or softened the language. Nonetheless, I have a few minor remarks for the authors to consider.

We sincerely appreciate the reviewer's positive and encouraging feedback. As suggested by the reviewer, we have addressed each comment individually below.

Furthermore, these studies typically employ advanced techniques such as high-resolution transmission electron microscopy (TEM), cryo-TEM, scanning electron microscopy (SEM), Raman spectroscopy, and X-ray scattering analysis, which, while informative, may introduce artifacts due to ex situ conditions and high-energy beam exposures.

Most of the cited studies were performed in situ and do not present any significant influence of artefacts. Hence, this statement should be revised.

We apologize for the oversight in our original statement. In response to the comment, we have revised the sentence to more accurately reflect the methodologies employed in the cited studies (**Lines 301-306**):

Other in situ techniques, including high-resolution transmission electron microscopy (TEM), cryo-TEM, and scanning electron microscopy (SEM), have provided valuable insights into homogeneous nucleation processes^{15, 24, 43, 60}. However, these techniques may face challenges in fully capturing in situ heterogeneous nucleation due to the complex interplay between nuclei and foreign surfaces under hydrated conditions.

Given recent observations of non-classical gypsum nucleation, it is striking to see how gypsum nucleation on foreign surfaces aligns the classical framework.

This is an ambiguous statement, as in many cases, nucleation data can be fitted using classical nucleation theory (CNT), even when the pathway is clearly non-classical. Furthermore, since there is currently no well-established non-CNT framework, the data cannot be accurately fitted using such a model, which complicates the direct comparison between both approaches.

We appreciate the reviewer's insightful feedback, which has helped us clarify and refine this statement in the manuscript (**Lines 323-326**).

Given that nucleation data can be fitted using CNT even when the pathway is non-classical, we recognize that fitting our data to CNT does not definitively imply a classical nucleation mechanism. To explore the possibility of an intermediate phase that may arise in non-classical pathways, we conducted...

By tracking the trajectories of CaSO₄ formed on the surface, a time-resolved plasmonic image sequence was captured (Supplementary Fig. 15a). Upon nucleation, a typical scattering pattern was clearly observed. As the time progressed, the plasmonic intensity increased (Supplementary Fig. 15b), indicating CaSO₄ nuclei growth. Notably, the monotonically increased plasmonic intensity excluded a phase change in CaSO₄. What if this phase change occurs gradually? This possibility actually seems to be supported by the RDF profiles, as stated by the authors: "However, when analyzing through the RDF profiles, it became evident that the inner ions within the clusters had started to organize into a crystalline structure. This evolving crystallinity is further supported by the comparison of RDF profiles over time (from 5 ns to 50 ns, Fig. R4), which demonstrates that the clusters were progressively moving toward a higher degree of order consistent with gypsum crystals."

Even at the end of the simulation run, the cluster structure is still significantly different from that of a gypsum crystal (Fig. R4), appearing to lie somewhere between a fully amorphous phase and fully crystalline phase.

Having pointed this out, I would like to emphasize that the data presented by the authors can be interpreted in two ways, depending on the perspective one adopts. Therefore, the data remains inconclusive at this stage, and in my opinion, it is premature to favor one model over the other. This is, of course, a perfectly valid conclusion and highlights the need for future research to resolve this issue.

We acknowledge the reviewer's concern that the monotonically increasing plasmonic intensity may not entirely rule out the possibility of a gradual phase change in CaSO₄. Your observation regarding the RDF profiles, which suggest evolving crystallinity, is well noted and underscores the complexity of the nucleation process.

In response to the reviewer's comments, we have revised our manuscript to soften our conclusions and to provide a more balanced perspective. The following revisions have been made:

Lines 335-336: *Notably, the monotonically increasing plasmonic intensity suggests that an abrupt phase transition in CaSO₄ is unlikely.*

Lines 343-344: *Therefore, no direct evidence of an intermediate phase was observed.*

Additionally, we calculated the great changes in the RDF of sulfur atom pairs of sulfate with time. To identify the specific structure of the clusters, the RDF diagrams of amorphous phases and perfect crystal of CaSO₄ were compared. As illustrated in Supplementary Fig. 16, the RDF peaks of amorphous calcium sulfate were broad and flattened, while the RDF peaks of crystalline calcium sulfate were sharp and well-defined, exhibiting a degree of long-range order. Simulations performed for the clusters over the period from 0 to 50 ns show that the behavior of the clusters was similar to that of a crystal phase.

The authors failed to mention that the simulations were conducted within the typical supersaturation range for homogeneous nucleation. Additionally, the distinction

between broad-flattened peaks and sharp, well-defined peaks appears to be rather subjective and lacks quantitative support.

We sincerely appreciate the reviewer's insightful comments. In response, we have added details regarding the ion concentrations used in the simulations, which were within the typical supersaturation range for homogeneous nucleation. We have also moderated our conclusions by using more tentative language.

Lines 377-343: *To identify the specific structure of the clusters, we compared the RDF diagrams of amorphous phases and perfect crystal of CaSO₄, using ion concentrations within the typical supersaturation range for homogeneous nucleation. As illustrated in Supplementary Fig. 16, the RDF peaks of amorphous calcium sulfate appeared broad and flattened, while the RDF peaks of crystalline calcium sulfate were sharp and well-defined, indicating a degree of long-range order. Simulations performed for the clusters over the period from 0 to 50 ns suggested that the behavior of the clusters was similar to that of a crystal phase.*

Lines 469-471: *To accelerate nucleation events within a feasible simulation timescale and minimize computational costs, the simulation was performed at elevated supersaturation levels.*

Therefore, the intermediate phase, commonly observed in non-classical nucleation process, could be excluded in this study.

This phrase is misleading because the absence of evidence does not equate to evidence of absence. The statement should be reformulated, for example, as: "In this study, no direct evidence of an intermediate phase was observed."

We agree with the reviewer's comment and have revised this statement accordingly (**Lines 343-344**): *Therefore, no direct evidence of an intermediate phase was observed.*

Response to Reviewer 3's Comments

Since the first review of this manuscript, it has evolved considerably. The authors' persistence in noteworthy and I highly appreciate this. I think that this stage the article can be accepted for publication.

We highly appreciate the reviewer's valuable time and efforts spent on our manuscript.